# Intra-species differences in population size shape life history and genome evolution

David Willemsen[1], Rongfeng Cui[1], Martin Reichard[2,3], Dario Riccardo Valenzano[1,4]*

[1]Max Planck Institute for Biology of Ageing, Cologne, Germany; [2]Czech Academy of Sciences, Institute of Vertebrate Biology, Brno, Czech Republic; [3]Department of Botany and Zoology, Faculty of Science, Masaryk University, Brno, Czech Republic; [4]CECAD, University of Cologne, Cologne, Germany

**Abstract** The evolutionary forces shaping life history divergence within species are largely unknown. Turquoise killifish display differences in lifespan among wild populations, representing an ideal natural experiment in evolution and diversification of life history. By combining genome sequencing and population genetics, we investigate the evolutionary forces shaping lifespan among wild turquoise killifish populations. We generate an improved reference genome assembly and identify genes under positive and purifying selection, as well as those evolving neutrally. Short-lived populations from the outer margin of the species range have small population size and accumulate deleterious mutations in genes significantly enriched in the WNT signaling pathway, neurodegeneration, cancer and the mTOR pathway. We propose that limited population size due to habitat fragmentation and repeated population bottlenecks, by increasing the genome-wide mutation load, exacerbates the effects of mutation accumulation and cumulatively contribute to the short adult lifespan.

*For correspondence:
dvalenzano@age.mpg.de

## Introduction

The extent to which drift and selection shape life history trait evolution across species in nature is a fundamental question in evolutionary biology. Variations in population size among natural populations is expected to affect the rate of accumulation of advantageous and slightly deleterious gene variants, hence impacting the relative contribution of selection and drift to genetic polymorphisms (*Lanfear et al., 2014*). Populations living in fragmented habitats, subjected to continuous and severe bottlenecks, are expected to undergo dramatic population size reduction and drift, which can significantly impact the accumulation of genetic polymorphisms in genes affecting important life history traits (*Nonaka et al., 2019*). The two main evolutionary theories of aging explain aging as the consequence of two fundamentally different processes. The mutation accumulation theory of aging (MA) attributes the evolution of aging to germline-encoded genetic variants accumulating in populations due to the age-dependent weakening of purifying selection, which becomes less efficient to remove from the gene pool gene variants that negatively impact fitness in late life (*Charlesworth, 2000*). The antagonistic pleiotropy (AP) theory of aging, instead, states that positive selection could favor gene variants that, while overall beneficial for individual fitness, may have detrimental effects in late life (*Charlesworth, 2000*; *Williams, 1957*). Although the two theories are not mutually exclusive and can both in principle explain the evolution of aging-related variants across species, their genetic traces in the genome should be distinguishable. In fact, while aging-determining gene variants occurring due to mutation accumulation evolve as nearly-neutral variants, aging-determining gene variants emerging via antagonistic pleiotropism evolve as positive selected variants.

Among vertebrates, killifish represent a unique system, as they repeatedly and independently colonized highly fragmented habitats, characterized by cycles of rainfalls and drought (*Furness, 2016*). While on the one hand intermittent precipitation and periodic drought pose strong selective pressures leading to the evolution of embryonic diapause, an adaptation that enables killifish to survive in absence of water (*Cellerino et al., 2016*; *Hu and Brunet, 2018*), on the other hand they cause habitat and population fragmentation, promoting inbreeding and genetic drift. The co-occurrence of strong selective pressure for early-life on the one hand and population size decline leading to genetic drift on the other hand characterizes life history evolution in African annual killifishes (*Cui et al., 2019*).

The turquoise killifish (*Nothobranchius furzeri*) is the shortest-lived vertebrate with a thoroughly documented post-embryonic life, which, in the shortest-lived strains, amounts to four months (*Cellerino et al., 2016*; *Hu and Brunet, 2018*; *Kim et al., 2016*; *Blazek, 2017*). Turquoise killifish has recently emerged as a powerful new laboratory model to study experimental biology of aging due to its short lifespan and to its wide range of aging-related changes, which include neoplasias (*Di Cicco et al., 2011*), decreased regenerative capacity (*Wendler et al., 2015*), cellular senescence (*Ahuja et al., 2019*; *Valenzano et al., 2006*), and loss of microbial diversity (*Smith et al., 2017*). At the same time, while sharing physiological adaptations that enable embryonic diapause and rapid sexual maturation, different wild turquoise killifish populations display differences in lifespan, both in the wild and in captivity (*Terzibasi et al., 2008*; *Valenzano et al., 2015*; *Vrtílek et al., 2018*), making this species an ideal evolutionary model to study the genetic basis underlying life history trait divergence within species.

Characterization of life history traits in wild-derived laboratory strains of turquoise killifish revealed that while different populations have similar rates of sexual maturation (*Blazek, 2017*), populations from arid regions exhibit the shortest lifespans, while populations from more semi-arid regions exhibit longer lifespans (*Blazek, 2017*; *Terzibasi et al., 2008*). Hence, speed of sexual maturation and adult lifespan appear to be independent in turquoise killifish populations. The evolutionary mechanisms responsible for the lifespan differences among turquoise killifish populations are not yet clearly understood. Mapping genetic loci associated with lifespan differences among turquoise killifish populations showed that adult survival has a complex genetic architecture (*Valenzano et al., 2015*; *Kirschner et al., 2012*). Here, combining genome sequencing and population genetics, we investigate to what extent genomic divergence in natural turquoise killifish populations that differ in lifespan is driven by adaptive or neutral evolution, compatible with either the antagonistic pleiotropy (AP) theory of aging or with the mutation accumulation (MA) theory of aging, respectively.

## Results

### Genome assembly improvement and gene annotation

To identify the genomic mechanism that led to the evolution of differences in lifespan between natural populations of the turquoise killifish (*Nothobranchius furzeri*), we combined the currently available reference genomes (*Valenzano et al., 2015*; *Reichwald et al., 2015*) into an improved reference turquoise killifish genome assembly. Due to the high repeat content, genome assembly from short reads required a highly integrated and multi-platform approach. We ran Allpaths-LG with all the available pair-end sequences, producing a combined assembly with a contig N50 of 7.8 kb, corresponding to a ~ 2 kb improvement from the previous versions. Two newly obtained 10X Genomics linked read libraries were used to correct and link scaffolds, resulting in a scaffold N50 of 1.5 Mb, that is a three-fold improvement from the best previous assembly. With the improved continuity, we assigned 92.2% of assembled bases to the 19 linkage groups using two RAD-tag maps (*Valenzano et al., 2015*). Gene content assessment using the BUSCO method improved "complete" BUSCOs from 91.43% (*Valenzano et al., 2015*) and 94.59% (*Reichwald et al., 2015*) to 95.20%. We mapped Genbank *N. furzeri* RefSeq RNA to the new assembly to predict gene models. The predicted gene model set is 96.1% for 'complete' BUSCOs. The overall size of repeated regions (masked regions) is 1.003 Gb, accounting for 66% of the entire genome, that is 20% higher than a previous estimate (*Reichwald et al., 2009*).

## Population genetics of natural turquoise killifish populations

Natural populations of turquoise killifish occur along an aridity gradient in Zimbabwe and Mozambique and populations from more arid regions are associated with shorter captive lifespan (*Blazek, 2017*; *Terzibasi et al., 2008*). A QTL study performed between short-lived and long-lived turquoise killifish populations showed a complex genetic architecture of lifespan (measured as age at death), with several genome-wide loci associated with lifespan differences among long-lived and short-lived populations (*Valenzano et al., 2015*). To further investigate the evolutionary forces shaping genetic differentiation in the loci associated with lifespan among wild turquoise killifish populations, we performed pooled whole-genome-sequencing (WGS) of killifish collected from four sampling sites within the natural turquoise killifish species distribution, which vary in altitude, annual precipitation and aridity (*Figure 1—figure supplement 1*, *Supplementary file 1A*). Population GNP is located within the Gonarezhou National Park at high altitude and in an arid climate (Koeppen-Geiger classification 'BWh', *Figure 1—figure supplement 1*), in a region at the outer edge of the turquoise killifish distribution (*Figure 1—figure supplement 1*; *Dorn et al., 2011*; *Bartáková et al., 2013*; *Bartáková et al., 2015*), which corresponds to the place of origin of the 'GRZ' laboratory strain, which has the shortest lifespan of all laboratory strains of turquoise killifish (*Terzibasi et al., 2008*; *Valenzano et al., 2015*). Population NF414 (MZCS 414) is located in an arid area in the center of the Chefu river drainage in Mozambique ('BWh', *Figure 1—figure supplement 1*; *Dorn et al., 2011*; *Bartáková et al., 2013*; *Bartáková et al., 2015*), and population NF303 (MZCS 303) is located in a semi-arid area in transition to more humid climate zones in the center of the Limpopo river drainage system (Koeppen-Geiger classification 'BSh', *Figure 1—figure supplement 1*; *Dorn et al., 2011*; *Bartáková et al., 2013*; *Bartáková et al., 2015*). Altitude among localities ranges from 344 m (GNP) to 68 m (NF303, *Figure 1—figure supplement 1a* and *Supplementary file 1A*). The temporary habitat of turquoise killifish populations differs in terms of altitude and aridity, as the ephemeral pools at higher altitude are drained earlier and persist for shorter time, while water bodies in habitats at lower altitude last longer (*Terzibasi et al., 2008*). Population GNP is therefore named 'dry', population NF414 is named 'intermediate' and population NF303 'wet' throughout the manuscript. The populations used in this study are from localities that belong to the same drainage system as those used in the previous QTL study and their relative position is included in *Figure 1—figure supplement 2*.

## High genetic differentiation and contrasting population demography in dry and wet populations

We asked whether populations from dry, intermediate and wet areas, corresponding to shorter and progressively longer lifespan, differ in genetic variability. We calculated genome-wide estimates of average pairwise difference ($\pi$) and genetic diversity ($\theta_{Watterson}$) based on 50kb-non-overlapping sliding windows using PoPoolation (*Kofler et al., 2011a*). We found that $\pi$ and $\theta_{Watterson}$ decrease from wet to dry population ($\theta_{Watterson\ GNP}$: 0.0011, $\theta_{Watterson\ NF414}$: 0.0036, $\theta_{Watterson\ NF303}$: 0.0072; $\pi_{GNP}$: 0.0009, $\pi_{NF414}$: 0.0031, and $\pi_{NF303}$: 0.0054). Hence, dry populations have overall smaller genetic diversity than populations from less dry regions. To infer the genetic distance between the populations, we computed the genome-wide pairwise genetic differentiation between populations using $F_{ST}$ (*Kofler et al., 2011b*). Overall, the genetic differentiation between populations ranged between 0.14 and 0.26 and was the highest between the more geographically distant population GNP (dry) and population NF303 (wet) (*Figure 1a*).

   Next, we inferred the demographic history of the populations using pairwise sequentially Markovian coalescent (PSMC) by resequencing at high-coverage single individuals for each population (*Schiffels and Durbin, 2014*). The population GNP (dry) experienced a strong population decline starting approximately 150 k generations ago, a result consistent for both the sequenced individuals from the two sampling sites (GNP-G1-3 and GNP-G4, *Figure 1a*). In contrast to the demographic history in GNP, we found indications for recent population expansions in populations from the center of the Chefu and Limpopo basins clades. Analysis of population NF414 (intermediate) (*Figure 1a*, NF414-Y and NF414-R) and NF303 (wet) (*Figure 1a*, blue line) shows population expansion until recent time (~50 k generations ago). To infer the effective population size ($N_e$) of the populations, we used the published mutational rate of 2.6321e−9 per base pair per generation for *Nothobranchius,* computed via dated phylogeny and $\theta_{Watterson}$ (*Cui et al., 2019*). In line with the

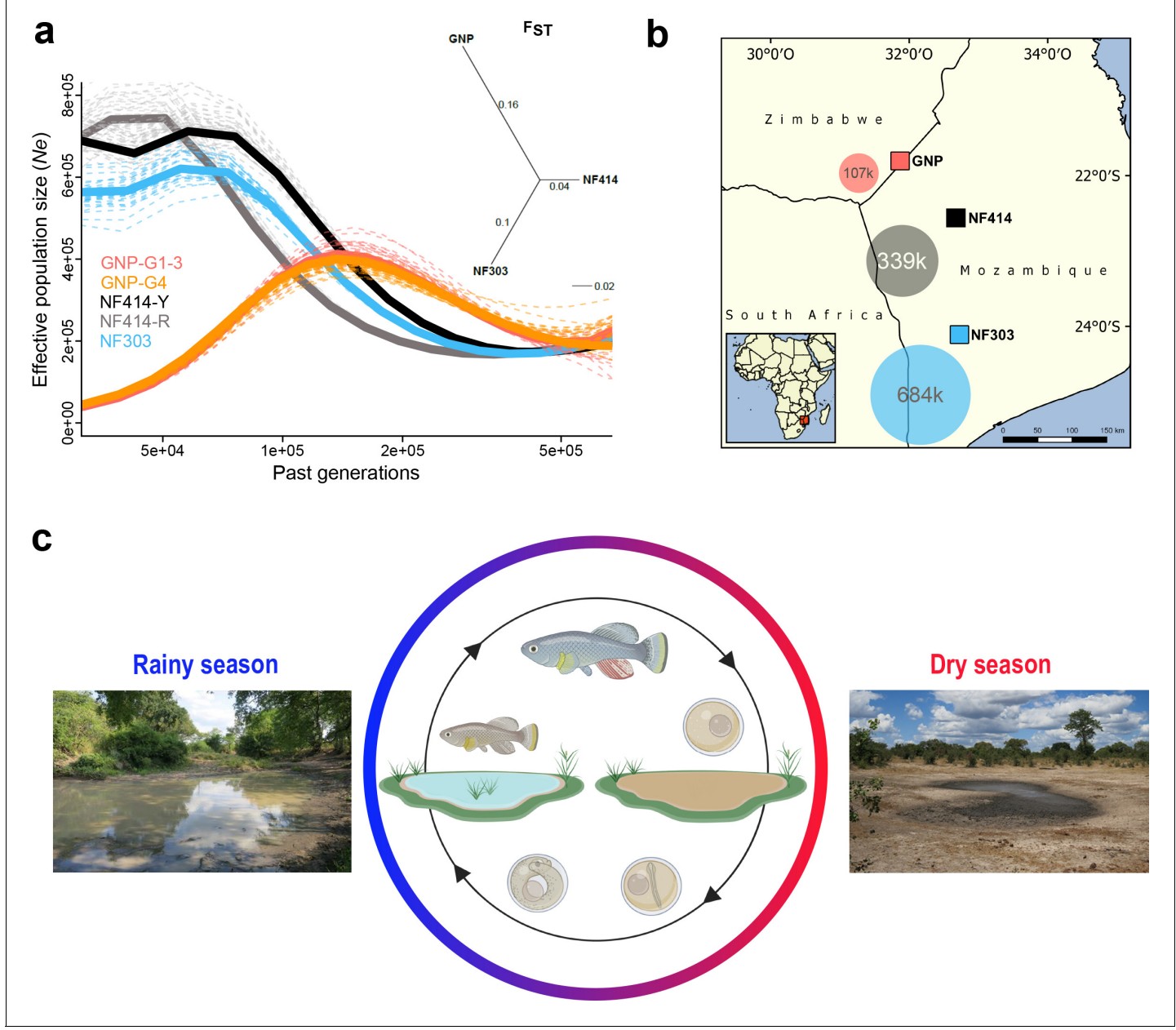

**Figure 1.** Demography and natural occurrence of turquoise killifish populations. (a) Inferred ancestral effective population size ($N_e$) (using PSMC') on y-axis and past generations on x-axis in GNP (red, orange), NF414 (black, grey) and NF303 (blue). Inset: unrooted neighbor joining tree based on pairwise genetic differentiation ($F_{ST}$) values. (b) Geographical locations of sampled natural population of turquoise killifish (*Nothobranchius furzeri*). The area of the colored circles represents the estimated effective population size ($N_e$) based on $\theta_{Watterson}$. (c) Natural environment of turquoise killifish and schematic of the annual life cycle. Figure 1 was partly made with Biorender.

The online version of this article includes the following figure supplement(s) for figure 1:

**Figure supplement 1.** Altitude, climate classification and genetic differentiation of studied samples.
**Figure supplement 2.** Map positions of the populations used in this study and of those used in the QTL study (*Valenzano et al., 2015*).

decrease in genetic diversity from wet to dry population, we found a decrease in $N_e$ estimates (107221.8, 338849.48 and 683693.25 for GNP, NF414 and NF303, respectively; *Figure 1b*). Hence, our findings show that dry populations from the outer edge of the species distribution show lower genetic diversity and smaller effective population size compared to population from intermediate and more wet regions.

## Genetic differentiation among turquoise killifish populations

To test whether regions underlying longevity QTL in turquoise killifish (*Valenzano et al., 2015*; *Kirschner et al., 2012*) display a genetic signature for positive or purifying selection in these wild populations, we took advantage of the improved turquoise killifish genome assembly and the newly sequenced wild turquoise killifish populations (*Figure 2*). The strongest QTL for lifespan differences among long-lived and short-lived populations mapped on the sex chromosome (*Valenzano et al., 2015*; *Kirschner et al., 2012*), in proximity to the sex determining locus (*Valenzano et al., 2015*).

To identify a genomic signature of strong selection, we performed an outlier approach based on the pairwise genetic differentiation index ($F_{ST}$). To find highly differentiated regions that may underlie positive selection in natural turquoise killifish populations, we scanned for regions with elevated genetic differentiation between pairs of populations, that is exceeding the 0.995 quantile of Z-transformed non-overlapping 50 kb sliding windows of $F_{ST}$. To find regions under purifying selection, we scanned for regions with lowered genetic differentiation among populations, that is below the 0.005 quantile of Z-transformed non-overlapping 50 kb sliding windows of $F_{ST}$ (*Supplementary file 1G*).

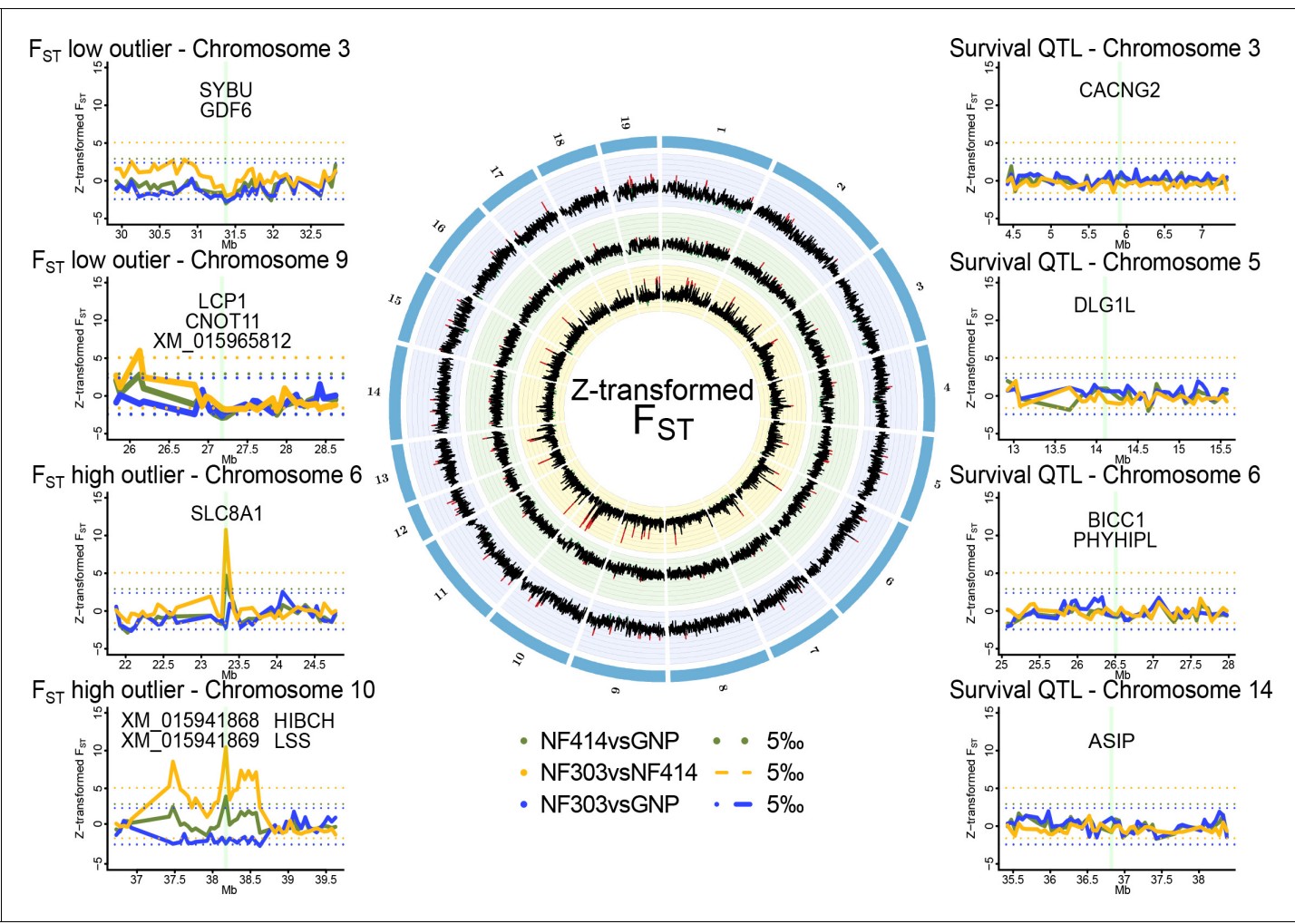

**Figure 2.** Genomic regions of high and low genetic divergence between pairs of turquoise killifish populations. Left) Genomic regions with high or low genetic differentiation between turquoise killifish populations identified with an $F_{ST}$ outlier approach. Z-transformed $F_{ST}$ values of all pairwise comparisons in solid lines, with 'NF303vsNF414' in yellow, 'NF303vsGNP' in blue, and 'NF414vsGNP' in green. The significance thresholds of upper and lower 5‰ are shown as dotted lines with same color coding. Center) Circos plot of Z-transformed $F_{ST}$ values between all pairwise comparisons with 'NF303vsNF414' in the inner circle (yellow), 'NF414vsGNP' in the middle circle (green), and 'NF303vsGNP' in the outer circle (blue). Right) Pairwise genetic differentiation based on $F_{ST}$ in the four main clusters associated with lifespan (QTL from *Valenzano et al., 2015*).

The outlier approach did not reveal clear signatures of positive or purifying selection based on genetic differentiation in the four main chromosomal clusters associated with lifespan in experimental strains of turquoise killifish (*Figure 2*).

We then analyzed genomic regions carrying signatures of positive and purifying selection in the natural turquoise killifish populations irrespective of the QTL regions (*Figure 2*). The $F_{ST}$ outlier approach led to the identification of several potential regions under strong selection between populations, in particular between the intermediate and wet populations (*Supplementary file 1D*) and only two between the dry and wet populations (*Supplementary file 1E*). Genes significantly different and within regions of larger genetic differentiation based on Z-transformed non-overlapping sliding windows of $F_{ST}$ were located on chromosomes 6 and 10. The region on chromosome six includes the gene *slc8a1*, which contains mutations with significant difference in allele frequencies between the wet and intermediate population (Fisher's exact test implemented in PoPoolation; adjusted p value < 0.001). The region on chromosome 10 contains four genes: XM_015941868, XM_015941869, *lss* and *hibch.* All genes under the major $F_{ST}$ peak on chromosome 10 showed significant difference in allele frequencies between the intermediate and wet population (Fisher's exact test; adjusted p value < 0.001) and additionally, *hibch* had significantly different allele frequencies between the dry and wet population (Fisher's exact test; adjusted p value < 0.001).

## Age-specific changes in genes with sequence divergence between populations

Genes under $F_{ST}$ peaks between populations that differ in lifespan, are not necessarily causally involved in lifespan differences between populations, as sequence differences could segregate in populations due to population structure and drift. However, to test whether the genes located in genomic regions that are significantly divergent between populations could be functionally involved in age-related phenotypes, we investigated whether gene expression in these genes varied as a function of age. Analyzing available turquoise killifish longitudinal RNA-Seq datasets generated in liver, brain and skin (*Baumgart et al., 2017*), we found that *hibch*, *lss* and *slc8a1* are differentially expressed between adult and old killifish (*Supplementary file 1J*, adjusted p value < 0.01). *hibch*, *lss and slc8a1* are involved in amino acid metabolism (*Ferdinandusse et al., 2013*), biosynthesis of cholesterol (*Huff and Telford, 2005*), and proton-mediated accelerated aging (*Osanai et al., 2018*), respectively. Gene XM_015956265 (ZBTB14) is the only gene that is an $F_{ST}$ outlier and that is differentially expressed in adult vs. old individuals between at least two populations in all tissues (liver, brain and skin). XM_015956265 encodes a transcriptional modulator with ubiquitous functions, ranging from activation of dopamine transporter to repression of myc, fmr1 and thymidine kinase promoters (*Orlov et al., 2007*). However, although genomic regions that have sequence divergence between turquoise killifish populations contain genes that are differentially expressed during aging in different tissues, whether any of these genes are causally involved in modulating aging-related changes between turquoise killifish wild populations still remains to be assessed. We could not find enrichment of significant differentially expressed genes within the $F_{ST}$ outlier regions (Fisher's exact test p value > 0.05).

## Genomic regions of low genetic differentiation among populations

Based on the outlier approach, we found two genomic regions with low genetic differentiation between all pairs of populations, suggesting strong purifying selection. The first region is located on the sex chromosome and contains the putative sex determining gene *gdf6* (*Reichwald et al., 2015*), which is hence conserved among these populations. This same region also contains *sybu*, a maternal-effect gene associated with the establishment of embryo polarity (*Nojima et al., 2010*). The second region under low genetic differentiation is located on chromosome nine and harbors the genes XM_015965812 (*abi2-like*), *cnot11* and *lcp1*, which are involved in phagocytosis (*Ulvila et al., 2011*), mRNA degradation (*Mauxion et al., 2013*) and cell motility (*Kell et al., 2018*), respectively. Signatures of low and high genetic differentiation between populations can be the result of purifying or positive selection. However, balancing selection, a mechanism that could maintain polymorphism above the expected genetic diversity, could also in part result in genetic differentiation (*Brandt et al., 2018*). To account for balancing selection, we compared the pairwise genetic

diversity (π) among populations and we could not find signatures of elevated genetic diversity within the investigated regions under strong selection.

Hence, we could not find a clear evidence of positive or purifying selection in correspondence with the survival QTL previously identified, suggesting that genomic regions associated with natural lifespan differences may have not evolved due to positive selection or have being maintained under purifying selection. However, we cannot exclude that we could not detect positive selection at the QTL regions due to statistical power or that the populations used in this study and those used for the QTL analysis had a different genetic architecture of lifespan.

## Evolutionary origin of the sex chromosome

Since we found reduced genetic differentiation among populations in the chromosomal region containing the putative sex-determining gene in the sex chromosome, we used synteny analysis and the new genome assembly to investigate the genomic events that led to evolution of this chromosomal region (*Figure 3*). We found that the structure of the turquoise killifish sex chromosome is compatible with a chromosomal translocation within an ancestral chromosome and a fusion event between two chromosomes. The translocation event within an ancestral chromosome corresponding to medaka´s chromosome 16 and platyfish´s linkage group three led to a repositioning of a chromosomal region containing the putative sex-determining gene *gdf6* (*Figure 3b*). The fusion of the translocated chromosome with a chromosome corresponding to medaka´s chromosome eight and platyfish´s linkage group 16, possibly led to the origin of turquoise killifish's sex chromosome. We could hence reconstruct a model for the origin of the turquoise killifish sex chromosome (*Figure 3c*), which parsimoniously places a translocation event before a fusion event. The occurrence of two major chromosomal rearrangements could have then contributed to suppressing recombination around the sex-determining region (*Valenzano et al., 2015*; *Valenzano et al., 2009*).

## Relaxed selection in turquoise killifish populations

Since we could not identify specific signatures of genetic differentiation in the genomic regions associated with longevity from previous QTL mapping, we asked whether other evolutionary forces than directional selection may underlie differences in survival among wild turquoise killifish populations. The difference in the recent and past demography between populations (*Figure 1*) led us to ask whether demography could have led to evolutionary changes on genome-wide scale between natural populations. For each population, we calculated the fraction of substitutions driven to fixation by positive selection since divergence from the outgroup species *Nothobranchius orthonotus* (NOR) using the asymptotic McDonald-Kreitman α (*Messer and Petrov, 2013*). The original McDonald-Kreitman α (which ranges from $-\infty$ to 1) was designed to calculated the rate of adaptation by comparing the polymorphisms (within species) and divergence (between species) at neutral and functional sites (*McDonald and Kreitman, 1991*). While McDonald-Kreitman α = 0 indicates neutrality, larger and positive values of α mean that a given population has an elevated proportion of genetic variants driven by natural selection, while negative values of α can be an indication of deleterious variants. The asymptotic McDonald-Kreitman α accounts for a range of derived allele frequencies, enabling to identify slightly deleterious mutations as those segregating at lower derived allele frequencies (*Messer and Petrov, 2013*). Variants at low derived allele frequency are either neutral (if they were beneficial they would have higher frequency) or are slightly deleterious. Hence, negative McDonald-Kreitman α values at low derived allele frequency bins likely reflect slightly deleterious gene variants. Additionally, negative α values at intermediate to higher frequency bins may indicate drift of deleterious variants. We set out to adopt this method to assess the genetic variants for each population, compared to two different outgroup species.

Using *Nothobranchius orthonotus* (NOR) as an outgroup, we inferred the fraction of positive selection by pooling all coding sites (*Figure 4a*). SNPs were called with the program SNAPE (*Raineri et al., 2012*), which specifically deals with pooled sequencing. We only included SNPs with a derived frequency between 0.05–0.95 and performed stringent filtering. The asymptotic McDonald-Kreitman α ranged from $-0.21$ to $-0.01$ in comparison to the very closely related sister species *N. orthonotus*, confirming limited genome-wide positive selection since divergence from *N. orthonotus* (*Figure 4a*). The population GNP, located in an arid region at higher altitude and associated with the shortest recorded lifespan, shows the lowest asymptotic McDonald-Kreitman α, as well as lower

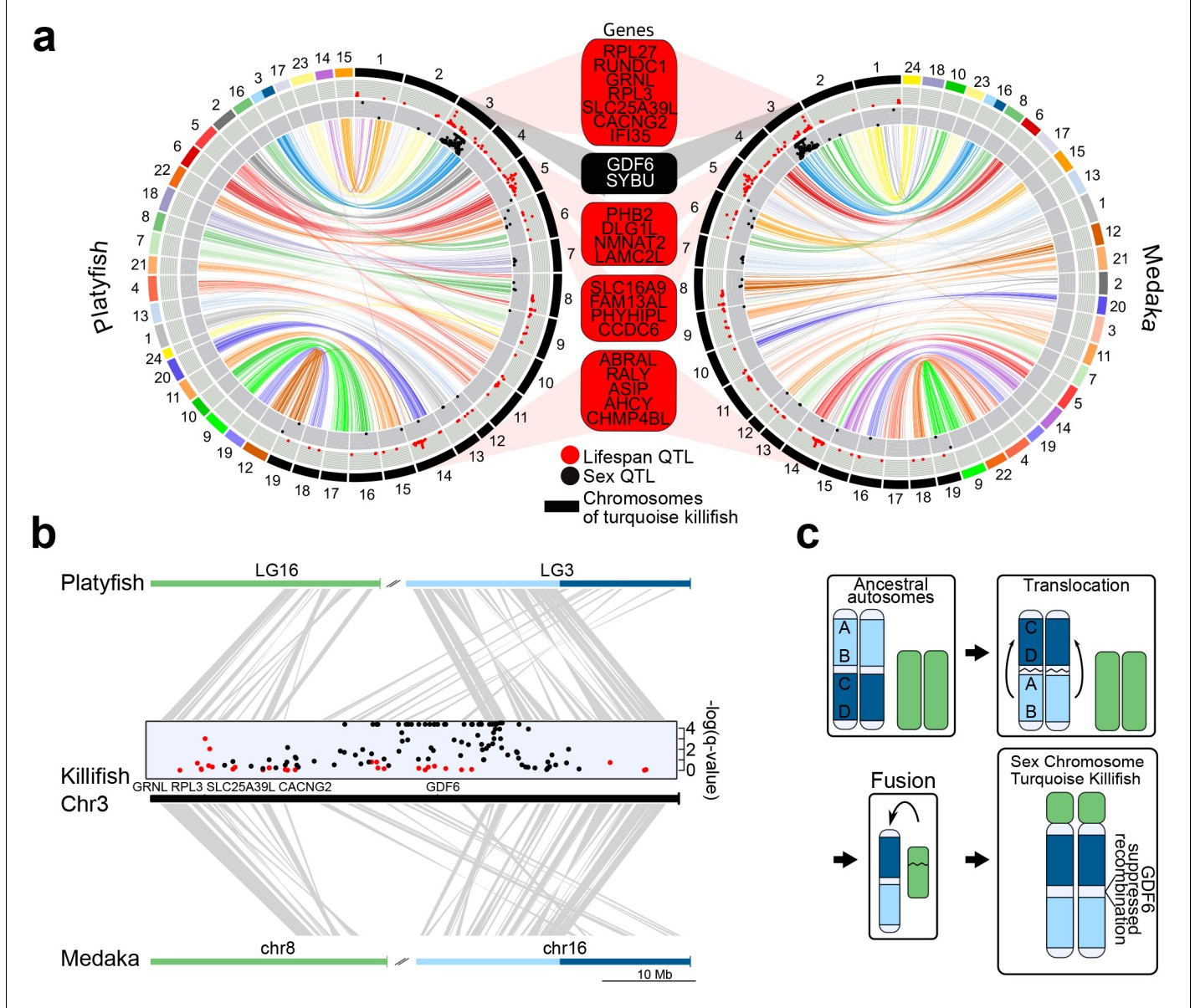

**Figure 3.** Synteny and sex chromosome evolution in turquoise killifish. (a) Synteny circos plots based on 1-to-1 orthologous gene location between the new turquoise killifish assembly (black chromosomes) and platyfish (*Xiphophorus maculatus*, colored chromosomes, left circos plot) and between the new turquoise killifish assembly (black chromosomes) and medaka (*Oryzias latipes*, colored chromosomes, right circos plot). Orthologous genes in concordant order are visualized as one syntenic block. Synteny regions are connected via color-coded ribbons, based on their chromosomal location in platyfish or medaka. If the direction of the syntenic sequence is inverted compared to the compared species, the ribbon is twisted. Outer data plot shows –log(q-value) of survival quantitative trait loci (QTL, ordinate value between 0 and 3.5, every value above 3.5 is visualized at 3.5 [*Valenzano et al., 2015*]) and the inner data plot shows –log(q-value) of the sex QTL (ordinate value between 0 and 3.5, every value above 3.5 is visualized at 3.5). Boxes between the two circos plots show genes within the peak regions of the four highest –log(q-value) of survival QTL on independent chromosomes (red box) and the highest association to sex (black box). (b) High resolution synteny map between the sex-chromosome of the turquoise killifish (Chr3) with platyfish chromosome 16 and 3 in the upper plot, and between the turquoise killifish and medaka chromosome 8 and 16 (lower plot). The middle plot shows the QTLs for survival and sex along the turquoise killifish sex chromosome. (c) Model of sex chromosome evolution in the turquoise killifish. A translocation event within one ancestral autosome led to the emergence of a chromosomal region harboring a new sex-determining-gene (SDG). The fusion of a second autosome led to the formation of the current structure of the turquoise killifish sex chromosome.

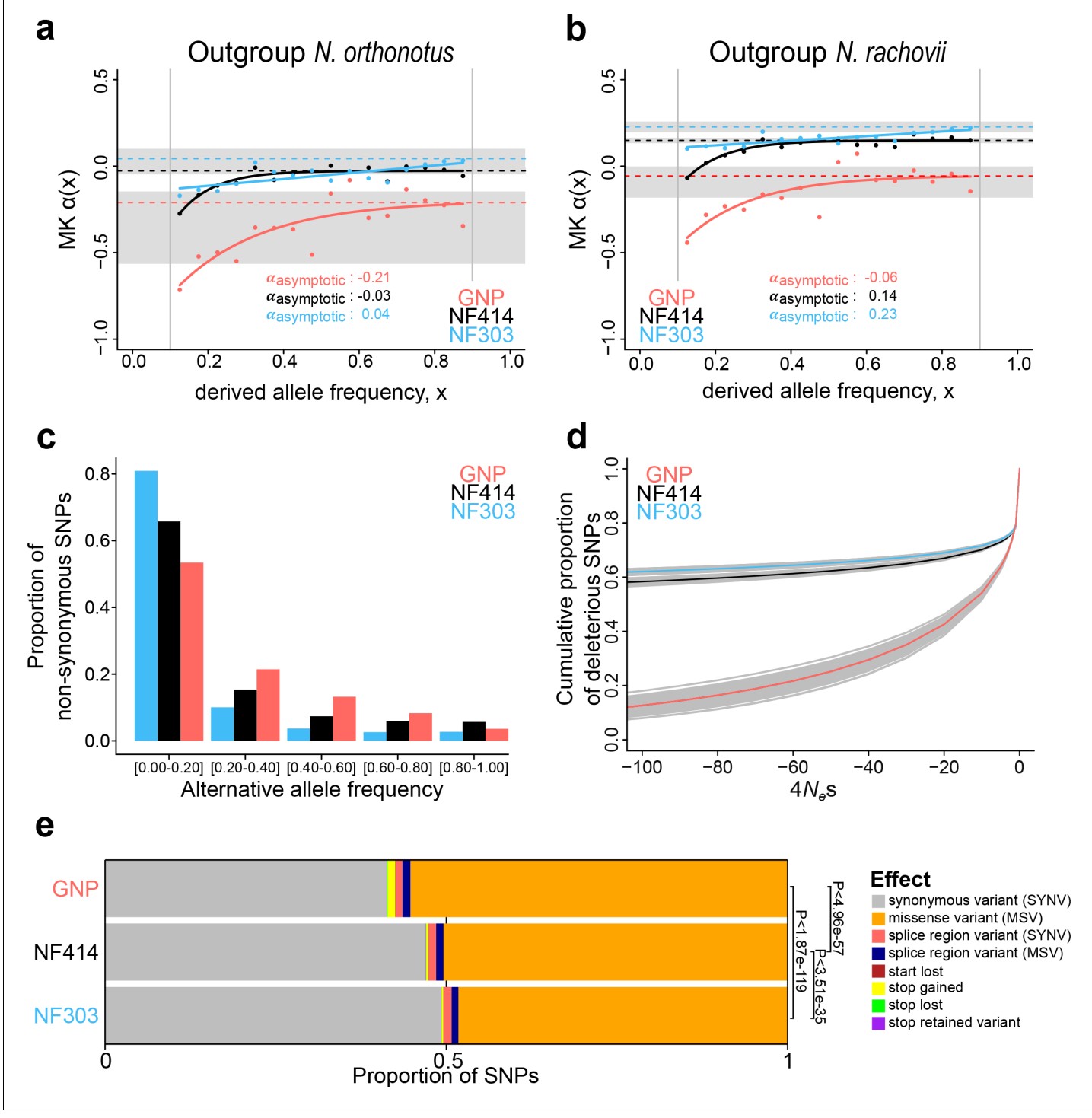

**Figure 4.** Genome-wide signatures of natural and relaxed selection in turquoise killifish populations. Asymptotic McDonald-Kreitman alpha (MK α) analysis based on derived frequency bins using as outgroups (**a**) *Nothobranchius orthonotus* and (**b**) *Nothobranchius rachovii*. Population GNP is shown in red, NF414 in black, and NF303 in blue. (**c**) Proportion of non-synonymous SNPs binned in allele frequencies of non-reference (alternative) alleles for GNP (red), NF414 (black) and NF303 (blue). (**d**) Negative distribution of fitness effects of populations GNP (red), NF414 (black) and NF303 (blue) with cumulative proportion of deleterious SNPs on y-axis and the compound measure of $4N_es$ on x-axis. (**e**) Proportion of different effect types of SNPs in coding sequences of all populations. The effect on amino acid sequence for each genetic variant is represented by colors (legend). Significance is based on ratio between synonymous effects to non-synonymous effects (significance based on Chi-square test).

The online version of this article includes the following figure supplement(s) for figure 4:

*Figure 4 continued on next page*

*Figure 4 continued*

**Figure supplement 1.** Simulated models with the inferred negative distribution of fitness effects.
**Figure supplement 2.** Mean Consurf score per variant based on derived frequency bins.

McDonald-Kreitman α values throughout all derived frequency bins, potentially suggesting a higher load of slightly deleterious mutations segregating in this population (*Figure 4a*). Using as an out-group species another annual killifish species, *Nothobranchius rachovii* (NRC), we confirmed the lowest asymptotic McDonald-Kreitman α value in the dry population GNP (*Figure 4b*). Additionally, using *Nothobranchius rachovii* (NRC) as outgroup species, the asymptotic McDonald-Kreitman α ranged from −0.06 to 0.23 among populations, indicating that more alleles were driven to fixation by positive selection in the ancestral lineage leading to *Nothobranchius furzeri* and *Nothobranchius orthonotus*. In particular, the wet population NF303 had the highest asymptotic McDonald-Kreitman α value (*Figure 4b*). Using both *N. orthonotus* and *N. rachovii* as outgroups, we found that the dry GNP population had the lowest McDonald-Kreitman α values at the low derived frequency bins, potentially consistent with a genome-wide accumulation of slightly deleterious mutations in these isolated populations.

## Estimating the distribution of fitness effect across populations

To directly estimate the fitness effect of gene variants associated with each population, we analyzed population-specific genetic polymorphisms to assign mutations as beneficial, neutral or detrimental, and determine the distribution of fitness effect (DFE) (*Tataru et al., 2017*) of new mutations. Consistently with the overall lower McDonald-Kreitman α values throughout all derived frequency bins, we found more new mutations assigned as the slightly deleterious category in the dry GNP population, compared to the other two populations (indicated by the higher number of deleterious SNPs in proximity to $4N_eS \sim 0$ in the GNP population, *Figure 4d*, *Supplementary file 1H*). To independently validate our findings, we ran a simulation using SLiM3, which recapitulated the population divergence from an ancestral population, followed with diverging population size as inferred from the PSMC' analysis (*Figure 4—figure supplement 1*). Analyzing the distribution of fitness effect in these simulated populations, we confirmed that populations with smaller effective population size have a higher proportion of new slightly deleterious variants, compared to larger populations, which have relatively more newly arising gene variants that are highly deleterious, indicating that purifying selection in the large population is more efficient in removing mutations with deleterious effects. To further infer the effect of the putative deleterious mutations on protein function, we used the new turquoise killifish genome assembly as a reference and adopted an approach that, by analyzing sequence polymorphism among populations, predicts functional consequences at the protein level (*Cingolani et al., 2012*). We found that the proportion of mutations causing a change in protein function is significantly larger in the GNP population compared to populations NF414 and NF303 (Chi-square test: $P_{GNP-NF303} <1.87e-119$, $P_{GNP-NF414} <4.96e-57$, $P_{NF303-NF414}< 3.51e-35$, *Figure 4e*). Additionally, the mutations with predicted deleterious effects on protein function reached also higher frequencies in the dry population GNP (*Figure 4c*).

## Distribution of mutations at conserved sites

To further investigate the impact of mutations on protein function, we calculated the Consurf (*Pupko et al., 2002*; *Mayrose et al., 2004*; *Glaser et al., 2003*; *Ashkenazy et al., 2016*) score, which determines the evolutionary constraint on an amino acid, based on sequence conservation. Mutations at amino acid positions with high Consurf score (i.e. otherwise highly conserved) are considered to be more deleterious. We found that the dry population GNP had a significantly higher mean Consurf score for mutations at non-synonymous sites in frequency bins from 5–20% up to 40–60%, compared to populations NF414 (intermediate) and NF303 (wet) (*Figure 4—figure supplement 2*). The mutations in the dry GNP population had significantly higher Consurf scores than the other populations using both outgroup species *N. orthonotus* and *N. rachovii* (Figure S3). Upon exclusion of potential mutations at neighboring sites (CMD: codons with multiple differences), CpG hypermutation and genes containing mutations with highly detrimental effect on protein function based on SnpEFF analysis, the dry population GNP had higher mean Consurf score at the low

frequency bin (*Figure 4—figure supplement 2*, *Supplementary file 1K-L*). To note, we also found a significantly higher average Consurf score at synonymous sites in GNP at low derived frequencies (*Figure 4—figure supplement 1*, *Supplementary file 1K-L*), possibly suggestive of an overall higher mutational rate in GNP.

## Relaxation of selection in age-related disease pathways

To further test whether populations from dry environments accumulated a higher load of deleterious gene variants, we computed the gene-wise direction of selection (DoS) (*Stoletzki and Eyre-Walker, 2011*) index, which measures the strength of selection based on the count of mutations in non-synonymous and synonymous sites. Indeed, we found support to the hypothesis that the dry, short-lived population GNP has significantly more slightly deleterious mutations segregating in the population, compared to the populations NF414 and NF303 (*Figure 5a*, Median NOR: GNP: −0.17, NF414: −0.02, NF303: −0.01; Median NRC: GNP: −0.14, NF414: 0.00, NF303: 0.00; Wilcoxon rank sum

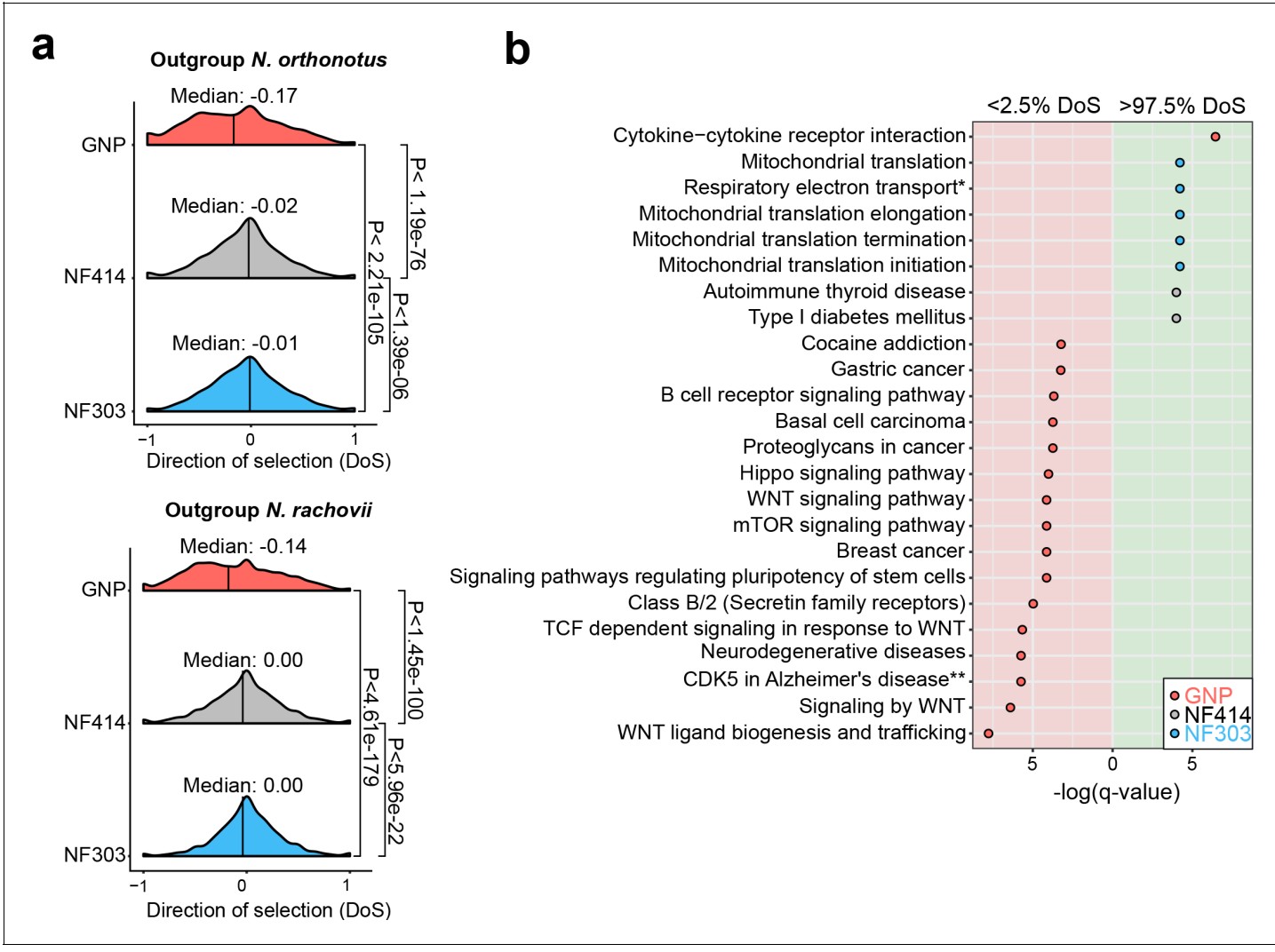

**Figure 5.** Pathway enrichment in genes under adaptive and neutral evolution in turquoise killifish populations. (**a**) Distribution of direction of selection (DoS) represented with median of distribution for population GNP (red), NF414 (grey) and NF303 (blue). Left panel shows DoS distribution computed using *Nothobranchius orthonotus* as outgroup and right panel shows DoS distribution computed using *Nothobranchius rachovii* as outgroup. Significance based on Wilcoxon-Rank-Sum test. (**b**) Pathway over-representation analysis of genes below the 2.5% level of gene-wise DoS values are shown with red background and above the 97.5% level of gene-wise DoS values are shown with green background. Only pathway terms with significance level of FDR corrected q-value <0.05 are shown (in -log(q-value)). Terms enriched in population GNP have red dots, enriched in population NF414 have black dots, and enriched in population NF303 have blue dots, respectively.

test: NOR: $P_{GNPNF303}$ <2.21e-105, $P_{GNP-NF414}$ <1.19e-76, $P_{NF303-NF414}$< 1.39e-06; NRC: $P_{GNP-NF303}$ <4.61e-179, $P_{GNP-NF414}$ <1.42e-100, $P_{NF303-NF414}$< 5.96e-22), indicating that purifying selection is relaxed in GNP. We calculated DoS in all populations using independently as outgroup species *N. orthonotus* and *N. rachovii* (*Figure 5a*).

To assess whether specific biological pathways were significantly more impacted by the accumulation of slightly deleterious mutations, we performed pathway overrepresentation analysis. We found a significant overrepresentation in the lower 2.5$^{th}$ DoS quantile (i.e. genes under relaxation of selection) in the GNP population for pathways associated with age-related diseases, including *gastric cancer, breast cancer, neurodegenerative disease, mTOR signaling* and *WNT signaling* (q-value <0.05, *Figure 5b*, *Supplementary file 1I*). Overall, relaxed selection in the dry GNP population affected accumulation of deleterious mutations in age-related and in the WNT pathway. Analyzing the pathways affected by genes within the upper 2.5$^{th}$ DoS values – corresponding to genes undergoing adaptive evolution – we found a significant enrichment for mitochondrial pathways – potentially compensatory (*Cui et al., 2019*) – in population NF303 (*Figure 5b*, *Supplementary file 1I*). Overall, our results show that differences in effective population size among wild turquoise killifish are associated with an extensive relaxation of purifying selection, significantly affecting genes involved in age-related diseases, and which could have cumulatively contributed to reducing individual survival.

## Discussion

The turquoise killifish (*Nothobranchius furzeri*) is the shortest-lived known vertebrate and while its natural populations show similar timing for sexual maturation, exhibit differences in lifespan along a cline of altitude and aridity in south-eastern Africa (*Blazek, 2017*; *Terzibasi et al., 2008*). Here we generate an improved genome assembly (NFZ v2.0) in turquoise killifish (*Nothobranchius furzeri*) and study the evolutionary forces shaping genome evolution among natural populations.

Using the new turquoise killifish genome assembly and synteny analysis with medaka and platyfish, we reconstructed the origin of the turquoise killifish sex chromosome, which appears to have evolved through two independent chromosomal events, that is a translocation and a fusion event.

Using the new genome assembly and pooled sequencing of natural turquoise killifish populations, we found that genetic differentiation among populations of the short-lived turquoise killifish is consistent with differences in demographic constraints. While we found that strong purifying selection maintains low genetic diversity among populations at genomic regions underlying key species-specific traits, such as in proximity to the sex-determining region, demography and genetic drift largely shape genome evolution, leading to relaxation of selection and the accumulation of deleterious mutations. We showed that isolated populations from an arid region, dwelling at higher altitude and characterized by shorter lifespan, experienced extensive population bottlenecking and a sharp decline in effective population size. Populations from dryer regions at higher altitudes experience genetic isolation and possibly steady decline in population size due to limited incoming gene flow and possibly more severe bottlenecks due to recent founder effect. However, populations from more wet regions likely undergo extensive gene flow, maintaining larger population size. We found that relaxation of selection in more drifted populations significantly affected the accumulation of deleterious gene variants in pathways associated with neurodegenerative diseases and WNT-signaling (*Figure 5*). While simple traits, such as male tail color and sex have a simple genetic architecture among turquoise killifish populations (*Valenzano et al., 2015*; *Valenzano et al., 2009*), we find that the complex genetic architecture of lifespan differences among killifish populations (*Valenzano et al., 2015*) is entirely compatible with genome-wide relaxation of selection. Additionally, the absence of genomic signature of positive selection in genomic regions underlying survival QTL in killifish suggest that, rather than directional selection, the neutral accumulation of deleterious mutations may be the evolutionary mechanism underlying survival differences among turquoise killifish populations, in line with the mutation accumulation theory of aging. The antagonistic pleiotropy theory of aging states that positive selection could lead to the fixation of gene variants that, while overall beneficial for fitness, could reduce survival and reproductive capacity in late life (*Williams, 1957*). However, the lack of genomic signatures of positive selection at the genomic regions underlying survival QTL in turquoise killifish rather suggests that the accumulation of deleterious mutations due to neutral drift may have played a key role in shaping genome and phenotype differences among natural turquoise killifish populations. One of the deductions of the antagonistic

pleiotropy theory is that a reduction in speed of maturation should be associated with increased life-span (*Williams, 1957*). However, different wild populations of turquoise killifish have similar time to sexual maturation and yet different lifespan (*Blazek, 2017*). Hence, the uncoupling of age of sexual maturation from adult lifespan in different turquoise killifish populations is more compatible with the mutation accumulation theory of aging. However, although we did not find evidence for it, our results do not exclude a priori the possibility that genes under strong selection may in part contribute to lifespan differences among different turquoise killifish populations, hence acting compatibly with the antagonistic pleiotropy theory of aging. However, historical fluctuations in the size of natural turquoise killifish populations, especially in isolated populations living in more arid and elevated habitats, weakened the strength of natural selection, ultimately contributing to increased load of deleterious gene variants, preferentially in genes associated with aging-related diseases and in the WNT pathway. We hypothesize that small effective population size leads to the accumulation of aging-causing mutations that together contribute to the genetic architecture of lifespan. Overall, our findings highlight the role of demographic constraints in shaping life history within species.

## Materials and methods

### Merging and improvement of the turquoise killifish genome assembly (BioProject ID: PRJNA599375)

#### 10x genomics read clouds

A single GRZ male individual was sacrificed with MS222 (Sigma-Aldrich, Steinheim, Germany). Blood was drawn from the heart and high molecular weight DNA was isolated with Qiagen MagAttract kit following manufacturer's instructions. Gemcode v2 DNA library generation was performed by Novogene (Beijing, China). Briefly, a proportion of the sample was run on a pulse field agarose gel to confirm high molecularity >100 kb. Based on a genome size estimate of 1.54 Gb (half of human genome), 0.6 ng of DNA was used to construct 2 Gemcode libraries, sequenced on two HiSeq X lanes to obtain a raw coverage of approximately 60X each. The reported input molecular length by SuperNova (*Weisenfeld et al., 2017*) was 118 kb for library 1 and 60.73 kb for library 2. Both libraries were used to correct and scaffold the Allpath-LG assembly (see below), and library one was also de novo assembled with the SuperNova assembler v.2 with default parameters. The SuperNova assembly totaled 802.6 Mb, with a contig N50 of 19.65 kb, scaffolded into 6.78 thousand scaffolds with an N50 of 3.83 Mb. Despite high continuity, however, the BUSCO (*Simão et al., 2015*) metrics are much lower than the Allpath-LG assemblies.

#### Nanopore long reads

DNA was extracted from a single GRZ male individual's muscle tissue by grinding in liquid nitrogen followed by phenol-chloroform extraction (Sigma). The rapid sequencing kit (SQK-RAD004) and the ligation kit (SQK-LSK108) were sued to prepare six libraries and were sequenced on 6 MinION flow cells (R9.4.1). These runs yielded a total of 3.3 Gb of sequences after trimming and correction by HALC (*Bao and Lan, 2017*). For correction, Allpath-LG contigs (see below) and short reads from the 10X genomic run were used.

#### Allpath-LG assembly

Two independent short read datasets were previous collected for the GRZ strain of *Nothobranchius furzeri*. Allpath-LG (*Gnerre et al., 2011*) was used on the pooled datasets. Together, 4 Illumina short read pair-end libraries with a fragment size distribution from 158 bp to 179 bp were used to construct the contigs (sequence coverage 191.9X, physical coverage 153.5X), and 22 pair-end and mate pair libraries distributed at 92 bp, 135 bp, 141 bp, 176 bp, 267 bp, 2 kb, 3 kb 5 kb and 10 kb were used for the scaffolding step (sequence coverage 135.7X, physical coverage 453.8X). The published BAC library ends (*Reichwald et al., 2015*) with an insert size of 112 kb were also included in the ALL-Paths-LG run (physical coverage 0.6X). The resulting assembly has a total contig length of 823,583,106 bp distributed in 151,307 contigs > 1 kb, with an N50 of 7.8 kb. The total scaffold length is 943,793,727 bp distributed in 7830 scaffolds with an N50 of 421 kb (with gaps). The resulting assembly was further scaffolded by ARCS v1.0 (*Coombe et al., 2018*) + LINKS v1.8.5 (*Warren et al., 2015*) with the following parameters: arcs -e 50000 c 3 r 0.05 s 98 and LINKS -m -d

4000 k 20 -e 0.1 l 3 -a 0.3 t 2 -o 0 -z 500 r -p 0.001 -x 0. This increased the scaffold N50 to 1.527 Mb. Next, scaffolds were assigned to the RAD-tag linkage map (*Valenzano et al., 2015*) collected from a previous study with Allmaps (*Tang et al., 2015*), using equal weight for the two independent mapping crosses. This procedure assigned 90.6% of the assembled bases in 1131 scaffolds to 19 linkage groups, in which 76.6% can be oriented. Misassemblies were corrected with the 10X genomic read cloud. Read clouds were mapped to the preliminary assembly with longranger v2.1.6 using default parameters, and a custom script was used to scan for sudden drops in barcode shares along the assembled linkage groups. The scaffolds were broken at the nearest gap of the drop in 10x barcodes. The same ARCS + LINKS pipeline was again run on the broken scaffolds, increasing the scaffold N50 to 1.823 Mb. Next, BESST_RNA (https://github.com/ksahlin/BESST_RNA) was used to further scaffold the assembly with RNASeq libraries, Allmaps was again used to assign the fixed scaffolds back to linkage groups, increasing the assignable bases to 92.2% (879Mb) with 80.3% (765Mb) with determined orientation. The assembly was again broken with longranger and reassigned to LG with Allmaps, and the scaffolds were further partitioned to linkage groups due to linkage of some left-over scaffolds with an assigned scaffold. Each partitioned scaffold groups were subjected to the ARCS + LINKS pipeline again, to constraint the previously unassigned scaffolds onto the same linkage group. Allmaps was run again on the improved scaffolds, resulting in 94.5% (903.4Mb) of bases assigned and 89.1% (852Mb) of bases oriented. Longranger was run again, visually checked and compared with the RADtag markers. Eleven mis-oriented positions were identified and corrected. Gaps were further patched by GMCloser (*Kosugi et al., 2015*) with ~2X of nanopore long reads corrected by HALC using BGI500 short PE reads with the following parameters: gmcloser —blast — long_read —lr_cov 2 l 100 -i 466 -d 13 min_subcon 1 min_gap_size 10 —iterate 2 —mq 1 c. The corrected long reads not mapped by GMCloser were assembled by CANU (*Koren et al., 2017*) into 7.9 Mb of sequences, which are likely unassigned repeats.

## Meta assembly

Five assemblies were integrated by MetAssembler (*Wences and Schatz, 2015*) in the following order (ranked by BUSCO scores) using a 20 kb mate pair library: 1) The improved Allpaths-LG assembly assigned to linkage groups produced in this study, 2) A previously published assembly with Allpaths-LG and optical map (*Reichwald et al., 2015*) 3) A previously published assembly using SGA (*Valenzano et al., 2015*), 4) The SuperNova assembly with only 10x Genomic reads and 5) Unassigned nanopore contigs from CANU. The final assembly NFZ v2.0 has 911.5 Mb of scaffolds assigned to linkage groups. Unassigned scaffolds summed up to 142.2 Mb, yielding a total assembly length of 1053.7 Mb, approximately 2/3 of the total genome size of 1.53 Gb. The final assembly has 95.2% complete and 2.24% missing BUSCOs.

## Mapping of NCBI Genbank gene annotations

RefSeq mRNAs for the GRZ strain (PRJNA314891, PRJEB5837) were downloaded from GenBank (*Sayers et al., 2019*), and aligned to the assembly with Exonerate (*Slater and Birney, 2005*). The RefSeq mRNAs have a BUSCO score of 98.0% complete, 0.9% missing. The mapped gene models resulted in a BUSCO score of 96.1% complete, 2.1% missing.

## Pseudogenome assembly generation

The pseudogenomes for *Nothobranchius orthonotus* and *Nothobranchius rachovii* were generated from sequencing data and the same method used in *Cui et al., 2019*. Briefly, the sequencing data were mapped to the NFZ v2.0 reference genome by BWA-mem v0.7.12 in PE mode (*Li, 2013*; *Li and Durbin, 2010*). PCR duplicates were marked with MarkDuplicates tool in the Picard (version 1.119, http://broadinstitute.github.io/picard/) package. Reads were realigned around INDELs with the IndelRealigner tool in GATK v3.4.46 (*McKenna et al., 2010*). Variants were called with SAMTOOLS v1.2 (*Li et al., 2009*) mpileup command, requiring a minimal mapping quality of 20 and a minimal base quality of 25. A pseudogenome assembly was generated by substituting reference bases with the alternative base in the reads. Uncovered regions, INDELs and sites with >2 alleles were masked as unknown 'N'. The allele with more supporting reads was chosen at biallelic sites.

## Mapping of longevity and sex quantitative trait loci

The quantitative trait loci (QTL) markers published in *Valenzano et al., 2015* were directly provided by Dario Riccardo Valenzano. In order to map the markers associated with longevity and sex, a reference database was created using BLAST (*Altschul et al., 1990*). The nucleotide database was created with the new reference genome of *N. furzeri* (NFZ v2.0). Subsequently, the QTL marker sequences were mapped to the database. Only markers with full support for the total length of 95 bp were considered as QTL markers.

## Synteny analysis

Synteny analysis was performed using orthologous information from *Cui et al., 2019* determined by the UPhO pipeline (*Ballesteros and Hormiga, 2016*). For this, the 1-to-1 orthologous gene positions of the new turquoise killifish reference genome (NFZ v2.0) were compared to two closely related teleost species, *Xiphophorus maculatus* and *Oryzias latipes*. Result were visualized using Circos (*Krzywinski et al., 2009*) for the genome-wide comparison and the *genoPlotR* package (*Guy et al., 2010*) in R for the sex chromosome synteny analysis. Synteny plots for orthologous chromosomes of *Xiphophorus maculatus* and *Oryzias latipes* were generated with Synteny DB (http://syntenydb.uoregon.edu) (*Catchen et al., 2009*).

## Koeppen-Geiger index and bioclimatic variables

The Koeppen-Geiger classification data was taken from *Peel et al., 2007* and the altitude, precipitation per month, and the bioclimatic variables were obtained from the Worldclim database (v2.0 [*Fick and Hijmans, 2017*]). The monthly evapotranspiration was obtained from *Trabucco and Zomer, 2019*. Aridity index was calculated based on the sum of monthly precipitation divided by sum of monthly evapotranspiration. Maps in *Figure 1—figure supplement 1* were generated with QGIS version 2.18.20 combined with GRASS version 7.4 (*Neteler et al., 2012*), the Koeppen-Geiger raster file, data from Natural Earth, and the river systems database from *Lehner and Verdin, 2006*.

## DNA isolation and pooled population sequencing

The ethanol preserved fin tissue was washed with 1X PBS before extraction. Fin tissue was digested with 10 µg/mL Proteinase K (Thermo Fisher) in 10 mM TRIS pH 8; 10 mM EDTA; 0.5 SDS at 50°C overnight. DNA was extracted with phenol-chloroform-isoamylalcohol (Sigma) followed by a washing step with chloroform (Sigma). Next, DNA was precipitated by adding 2.5 vol of chilled 100% ethanol and 0.26 vol of 7.5M Ammonium Acetate (Sigma) at −20°C overnight. DNA was collected via centrifugation at 4°C at 12000 rpm for 20 min. After a final washing step with 70% ice-cold ethanol and air drying, DNA was eluted in 30 µl of nuclease-free water. DNA quality was checked on one agarose gels stained with RotiSafe (Roth) and a UV-VIS spectrometer (Nanodrop2000c, Thermo Scientific). DNA concentration was measured with Qubit fluorometer (BR dsDNA Assay Kit, Invitrogen). For each population, the DNA of the individuals were pooled at equimolar contribution (GNP_G1_3, GNP_G4 N = 29; NF414, NF303 N = 30). DNA pools were given to the Cologne Center of Genomic (CCG, Cologne, Germany) for library preparation. The total amount of DNA provided to the sequencing facility was 3.2 µg per pooled population sample. Libraries were sequenced with 150 bp x two paired-ends on the HiSeq4000. Sequencing of pooled samples resulted in a range of 419–517 million paired-end reads for each population (*Supplementary file 1B*).

## Mapping of pooled sequencing reads

Raw sequencing reads were trimmed using Trimmomatic-0.32 (ILLUMINACLIP:illumina-adaptors. fa:3:7:7:1:true, LEADING:20, TRAILING:20, SLIDINGWINDOW:4:20, MINLEN:50 [*Bolger et al., 2014*]). Data files were inspected with FastQC (version 0.11.22, https://www.bioinformatics.babraham.ac.uk/projects/fastqc/). Trimmed reads were subsequently mapped to the reference genome with BWA-MEM v0.7.12 (*Li, 2013*; *Li and Durbin, 2010*). The SAM output was converted into BAM format, sorted, and indexed via SAMTOOLS v1.3.1 (*Li et al., 2009*). Filtering and realignment was conducted with PICARD v1.119 (http://broadinstitute.github.io/picard/) and GATK (*McKenna et al., 2010*). Briefly, the reads were relabeled, sorted, and indexed with AddOrReplaceReadGroups. Duplicated reads were marked with the PICARD feature MarkDuplicates and reads were realigned with first creating a target list with RealignerTargetCreator, second by IndelRealigner from the

GATK suite. Resulting reads were again sorted and indexed with SAMTOOLS. For population genetic bioinformatics analyses the BAM files of the pooled populations were converted into the required MPILEUP format via the SAMTOOLS mpileup command. Low quality reads were excluded by setting a minimum mapping quality of 20 and a minimum base quality of 20. Further, possible insertion and deletions (INDELs) were identified with *identifygenomic-indel-regions.pl* script from the PoPoolation package (*Kofler et al., 2011a*) and were subsequently removed via the *filter-pileup-by-gtf.pl* script (*Kofler et al., 2011a*). Coding sequence positions that were identified to be putative ambiguous were removed by providing the *filter-pileup-by-gtf.pl* script a custom modified GTF file with the corresponding coordinates. After adapter and quality filtering, mapping to the newly assembled reference genome resulted in mean genome coverage of 35x, 39x, and 47x for the population NF303, NF414, and GNP, respectively (*Supplementary file 1B*).

## Merging sequencing reads of populations from the Gonarezhou National Park

Population GNP consists of two sampling sites (GNP-G1_3, GNP-G4) with very low genetic differentiation (*Figure 1—figure supplement 1c*, *Supplementary file 1C*). Sequencing reads of the two populations from the Gonarezhou National Park (GNP) were combined used the SAMTOOLS 'merge' command. The populations GNP-G1-3 and GNP-G4 were merged together and this population was subsequently denoted as GNP.

## Estimating genetic diversity

Genetic diversity in the populations was estimated by calculating the nucleotide diversity π (*Nei and Li, 1979*) and Wattersons's estimator θ (*Watterson, 1975*). Calculation of π and θ was done with a sliding window approach by using the *Variance-sliding.pl* script from the PoPoolation program (*Kofler et al., 2011a*). Non-overlapping windows with a length of 50 kb with a minimum count of two per SNP, minimum quality of 20 and the population specific haploid pool size were used (GNP = 116; NF414 = 60; NF303 = 60). Low covered regions that fall below half the mean coverage of each population were excluded (GNP = 23; NF414 = 19; NF303 = 18), as well as regions that exceed a two times higher coverage than the mean coverage (GNP = 94; NF414 = 77; NF303 = 70). The upper threshold is set to avoid regions with possible wrong assemblies. Mean coverage was estimated on filtered MPILEUP files. Each window had to be at least covered to 30% to be included in the estimation.

## Estimation of effective population size

Wattersons's estimator of θ (*Watterson, 1975*) is referred to as the population mutation rate. The estimate is a compound parameter that is calculated as the product of the effective population size ($N_e$), the ploidy (2 p, with p is ploidy) and the mutational rate μ ($\theta = 2pN_e\mu$). Therefore, $N_e$ can be obtained when θ, the ploidy and the mutational rate μ are known. The turquoise killifish is a diploid organism with a mutational rate of 2.6321e−nine per base pair per generation (assuming one generation per year in killifish *Cui et al., 2019* and θ estimates were obtained with PoPoolation (see previous Section) (*Kofler et al., 2011a*).

## Estimating population differentiation index $F_{ST}$

The filtered and realigned BAM files of each population were merged into a single pileup file with SAMTOOLS mpileup, with a minimum mapping quality and a minimum base quality of 20. The pileup was synchronized using the *mpileup2sync.jar* script from the PoPoolation2 program (*Kofler et al., 2011b*). Insertions and deletions were identified and removed with the *identify-indel-regions.pl* and *filter-sync-by-gtf.pl* scripts of PoPoolation2 (*Kofler et al., 2011b*). Again, coding sequence positions that were identified to be putative ambiguous were removed by providing the *filter-pileup-by-gtf.pl* script a custom modified GTF file with the corresponding coordinates. Further a synchronized pileup file for genes only were generated by providing a GTF file with genes coordinates to the *create-genewise-sync.pl* from PoPoolation2 (*Kofler et al., 2011b*). $F_{ST}$ was calculated for each pairwise comparison (GNP vs NF303, GNP vs NF414, NF414 vs NF303) in a genome-wide approach using non-overlapping sliding windows of 50 kb with a minimum count of four per SNP, a minimum coverage of 20, a maximum coverage of 94 for GNP, 77 for NF414, and 70 for NF303 and

the corresponding pool size of each population (N = 116; 60; 60). Each sliding window had to be at least covered to 30% to be included in the estimation. The same thresholds, except the minimum covered fraction, with different sliding window sizes were used to calculate the gene-wise $F_{ST}$ for the complete gene body (window-size of 2000000, step-size of 2000000) and single SNPs within genes (window-size of 1, step-size of 1). The non-informative positions were excluded from the output. Significance of allele differences per base-pair within the gene-coordinates were calculated with the fisher´s exact test implemented in *the fisher-test.pl* script of PoPoolation2 (*Kofler et al., 2011b*). Calculation of unrooted neighbor joining tree based on the genome-wide pairwise $F_{ST}$ averages was performed with the ape package in R (*Paradis et al., 2004*).

## Detecting signatures of selection based on $F_{ST}$ outliers

For $F_{ST}$-outlier detection, the pairwise 50kb-window $F_{ST}$-values for each comparison were Z-transformed ($Z_{FST}$). Next, regions potentially under strong selection were identified by applying an outlier approach. Outliers were identified as non-overlapping windows of 50 kb within the 0.5% of lowest and highest genetic differentiation per comparison. To reduce the number of false-positive results, the outlier threshold was chosen at 0.5% highest and lowest percentile of each pairwise genetic differentiation (*Pruisscher et al., 2018*; *Guo et al., 2016*). To find candidate genes within windows of highest differentiation, a total of three selection criteria were used. First, the window-based $Z_{FST}$ value had to be above the 99.5th percentile of pairwise genetic differentiation. Second, the gene $F_{ST}$ value had to be above the 99.5th percentile of pairwise genetic differentiation and last, the gene needed to include at least one SNP with significant differentiation based on Fisher's exact test (calculated with PoPoolation2 [*Kofler et al., 2011b*]; p<0.001, Benjamini-Hochberg corrected P-values [*Benjamini and Hochberg, 1995*]).

## Identifying polymorphic sites

SNP calling was performed with Snape (*Raineri et al., 2012*). The program requires information of the prior nucleotide diversity θ. Hence, the initial values of nucleotide diversity obtained with PoPoolation were used. Snape was run with folded spectrum and prior type informative. As Snape requires the MPILEUP format, the previously generated MPILEUP files were used. SNP calling was separately performed on coding and non-coding parts of the genome. Therefore, each population MPILEUP file was filtered by coding sequence position with the *filter-pileup-by-gtf.pl* script of PoPoolation. For coding sequences, the `-keep-mode` was set to retain all coding sequences. The non-coding sequences were obtained by using the default option and thus discarding the coding sequences from the MPILEUP file. Snape produces a posterior probability of segregation for each position. The posterior probability of segregation was used to filter low-confidence SNPs and indicated in the specific section.

## Divergence and polymorphisms in 0-fold and 4-fold sites

Polarization of synonymous sites (four-fold degenerated sites) and non-synonymous sites (zero-fold degenerated sites) was done using the pseudogenomes of outgroups *Nothobranchius orthonotus* and *Nothobranchius rachovii*. For each population the genomic information of the respective pseudogenome was extracted with bedtools getfasta command (*Quinlan, 2014*; *Quinlan and Hall, 2010*) and the derived allele frequency of every position was inferred with a custom R script. Briefly, only sites with the bases A, G, T or C in the outgroup pseudogenome were included and checked whether the position has an alternative allele in each of the investigated populations. Positions with an alternative allele present in the population data were treated as possible divergent or polymorphic sites. The derived frequency was determined as frequency of the allele not present in the outgroup. Occasions with an alternate allele present in the population data were treated as possible divergent or polymorphic sites. Divergent sites are positions in the genome were the outgroup allele is different from the allele present in the population. Polymorphic sites are sites in the genome that have more than one allele segregating in the population. Only biallelic polymorphic sites were used in this analysis. The DAF was determined as frequency of the allele not shared with the respective outgroups. In general, positions with only one supporting read for an allele were treated as monomorphic sites. SNPs with a DAF < 5% or>95% were treated as fixed mutations. Further filtering was done based on the threshold of the posterior probability of >0.9 calculated with Snape (see previous

subsection), combined with a minimum and maximum coverage threshold per population (GNP: 24, 94; NF414:19, 77; NF303: 18, 70).

## Asymptotic McDonald-Kreitman α

The rate of substitutions that were driven to fixation by positive selection was evaluated with an improved method based on the McDonald-Kreitman test (*McDonald and Kreitman, 1991*). The test assumes that the proportion of non-synonymous mutations that are neutral has the same fixation rate as synonymous mutations. Therefore, under neutrality the ratio between non-synonymous to synonymous substitutions (Dn/Ds) between species is equal to the ratio of non-synonymous to synonymous polymorphisms within species (Pn/Ps). If positive selection takes place, the ratio between non-synonymous to synonymous substitutions between species is larger than the ratio of non-synonymous to synonymous polymorphism within species (*McDonald and Kreitman, 1991*). The concept behind this is that the selected variant reaches fixation in a shorter time than by random drift. Therefore, the selected variant increases Dn, not Pn. The proportion of non-synonymous substitutions that were fixed by positive selection (α) was estimated with an extension of the McDonald-Kreitman test (*Smith and Eyre-Walker, 2002*). Due to the presence of slightly deleterious mutations the estimate of α can be underestimated. For this reason, the method used by Messer and Petrov was implemented to calculate α as a function of the derived allele frequency x (*Messer and Petrov, 2013*; *Haller and Messer, 2017*). With this method the true value of α can be inferred as the asymptote of the function of α. Additionally, the value of α(x) for low derived frequencies should give an estimate of the number of slightly deleterious mutations that segregate in the population.

## Direction of selection (DoS)

To further investigate the signature of selection, the direction of selection (DoS) index for every gene was calculated (*Stoletzki and Eyre-Walker, 2011*). DoS standardizes α to a value between −1 and 1. A positive value of DoS indicates adaptive evolution (positive selection) and a negative value indicates the segregation of slightly deleterious alleles, therefore weaker purifying selection (*Stoletzki and Eyre-Walker, 2011*). This ratio is undefined for genes without any information about polymorphic or substituted sites. Therefore, only genes with at least one polymorphic and one substituted site were included.

## Inference of distribution of fitness effects

The distribution of fitness effects (DFE) was inferred using the program polyDFE2.0 (*Tataru et al., 2017*). For this analysis the unfolded site frequency spectra (SFS) of non-synonymous (0-fold) and synonymous sites (4-fold) were projected into 10 chromosomes for each population. Information about the fixed derived sites was included in this analysis (using *Nothobranchius orthonotus*). Poly-DFE2.0 estimates either the full DFE, containing deleterious, neutral and beneficial mutations, or only the deleterious DFE. The best model for each population was obtained using a model testing approach with three different models implemented in PolyDFE2.0 (Model A, B, C). Due to possible biases from erroneous polarization or unknown demography, runs accounting for polarization errors and demography (+eps, +r) were included. Initial parameters were automatically estimated with the –e option, as recommended. To ensure that the parameter space is explored thoroughly, the basin hopping option was applied with a maximum of 500 iterations (-b). The best model for each population was chosen based on the Akaike Information Criterion (AIC). Confidence intervals were generated by running 200 bootstrap datasets with the same parameters used to infer the best model.

## Simulations of population demography and the distribution of fitness effects

We performed simulations with SLiM (version 3.3) (*Haller and Messer, 2019*), using demographic parameters from the PSMC' analysis. We simulated four models with two populations that diverged from the same ancestral population and are identical in all population genetic parameters. We adapted the simulation from *Rousselle et al., 2018* and simulated 1500 coding sequences of 500 bp length, separated by non-coding regions and re-scaled the population genetic parameter to an initial population size of 10 k (mutation rate of 5.468928e-08, recombination rate of 6.604764e-06). The neutral mutations were initialized in ~1/3 of the coding region and deleterious mutations were

initialized in ~2/3 of the coding region, resembling the structure of a codon. Deleterious mutations were simulated with a selection coefficient s, a measure of relative fitness, drawn from the same reflected gamma distribution (mean of −2.5 and shape of 0.35) and a dominance coefficient of 0.1. All models start with an initial burn-in phase of 50000 generations to generate genetic diversity and a stable population. The models vary in either having immediate population size changes followed by constant population size (models A, B) or having exponential growth (models C, D). We further distinguished between a split directly after the burn-in phase (models B, D) or at a later timepoint, following the PSMC' analysis interpretation (models A, C). The scripts are available on https://github.com/valenzano-lab/simulation_DFE. We performed 25 replicates per model and used a sample size of 30 diploid individuals to retrieve the fixed and segregating mutations. The synonymous and non-synonymous site-frequency-spectra were built by projecting the retrieved mutations into 10 chromosomes, comparable to the site-frequency-spectra generated for the observed data of the turquoise killifish populations and subsequently used as the input for PolyDFE2.0. Model choice of the negative distribution of fitness effects was performed similar to the initial analysis as described previously, except we did not include the ancestral misidentification error.

## Variant annotation

Classification of changes in the coding-sequence (CDS) was done with the variant annotator SnpEFF (*Cingolani et al., 2012*). The new genome of *Nothobranchius furzeri* (NFZ v2.0) was implemented to the SnpEFF pipeline. Subsequently, a database for variant annotation with the genome NFZ v2.0 FASTA file and the annotation GTF file was generated. For variant annotation the population specific synonymous and non-synonymous sites with a change in respect to the reference genome NFZ v2.0 were used to infer the impact of these sites. The possible annotation impact classes were low, moderate, and high. SNPs with a frequency below 5% or above 95% were excluded for this analysis. To be consistent with the analysis of the distribution of fitness effects, only positions also found to be present in the *N. orthonotus* pseudogenome were considered. Positions with warnings in the variant annotation were removed.

## Consurf analysis

The Consurf score was calculated accordingly to the method used in *Cui et al., 2019*. We used the Consurf (*Pupko et al., 2002*; *Mayrose et al., 2004*; *Glaser et al., 2003*; *Ashkenazy et al., 2016*) package to assign each AA a conservation score based on the evolutionary rate in homologs of other vertebrates. Consurf scores were estimated for 12575 genes of *N. furzeri* and synonymous and non-synonymous genomic positions were matched with the derived allele frequency of *N. orthonotus* and *N. rachovii*, respectively. The derived frequencies were binned in five bins and we used pairwise Wilcoxon rank sum test to assess significance after correcting for multiple testing (Benjamini and Hochberg adjustment) between each subsequent bin per population and matching bins between populations.

## Over-representation analysis

Gene ontology (GO) and pathway overrepresentation analysis was performed with the online tool ConsensusPathDB (http://cpdb.molgen.mpg.de; version34) (*Herwig et al., 2016*) using 'KEGG' and 'REACTOME' databases. Briefly, each gene present in the outlier list was provided with an ENSEMBL human gene identifier (*Zerbino et al., 2018*), if available, and entered as the target list into the user interface. All genes included in the analysis and with available human ENSEMBL identifier were used as the background gene list. ConsensusPathDB maps the entries to the databases and calculates the enrichment score for each entity by comparing the proportion of target genes in the entity over the proportion of background genes in the entity. For each of the enrichment a P-value is calculated based on a hyper geometric model and is corrected for multiple testing using the false discovery rate (FDR). Only GO terms and pathways with more than two genes were included. Overrepresentation analysis was performed on genes falling below the 2.5th percentile or above the 97.5th percentile thresholds. The percentiles for either $F_{ST}$ or DoS values were calculated with the quantile() function in R.

## Statistical analysis and data processing

Statistical analyses were performed using R studio version 1.0.136 (R version 3.3.2 [*RStudio, 2015*]) on a local computer and R studio version 1.1.456 (R version 3.5.1) in a cluster environment at the Max-Planck-Institute for Biology of Ageing (Cologne). Unless otherwise stated, the functions t.test() and wilcox.test() in R have been used to evaluate statistical significance. To generate a pipeline for data processing we used Snakemake (*Köster and Rahmann, 2012*). Figure style was modified using Inkscape version 0.92.4. For circular visualization of genomic data we used Circos (*Krzywinski et al., 2009*).

## Inference of demographic population history with individual resequencing data

To infer the demographic history, we performed whole genome re-sequencing of single individuals from all populations resulting in mean genome coverage between 13-21x (*Supplementary file 1B*). Demographic history was inferred from single individual sequencing data using Pairwise Sequential Markovian Coalescence (PSMC' mode from MSMC2 [*Schiffels and Durbin, 2014*]). Re-sequencing of single individuals was performed with the DNA of single individuals extracted for the pooled sequencing for each examined population. The Illumina short-insert library was constructed based on a published protocol (*Rowan et al., 2015*). Extracted DNA (500 ng) was digested with fragmentase (New England Biolabs) for 20 min at 37°C, followed by end-repair and A-tailing (1.0 μl NEB End-repair buffer, 0.5 μl Klenow fragment, 0.5 μl Taq.Polymerase, 0.2 μl T4 polynucleotide kinase, 10 μl reaction volume, 30 min at 25°C, 30 min at 75°C) and adapter ligation (NEB Quick ligase buffer 12.5 μl, Quick ligase 0.5 μl, 1 μl adapter P1 (D50X), 1 μl adapter P2 (universal), 5 μM each; 20 min at 20° C, 25 μl reaction volume). Next, ligation mix was diluted to 50 μl and used 0.583:1 vol of home-brewed SPRI beads (SPRI binding buffer: 2.5M NaCL, 20 mM PEG 8000, 10 mM Tris-HCL, 1 mM EDTA,ph = 8, 1 mL TE-washed SpeedMag beads GE Healthcare, 65152105050250 per 100 mL buffer) for purification. The ligation products were amplified with 9 PCR cycles using KAPA Hifi kit (Roche, P5 universal primer and P7 indexed primer D7XX). The samples were pooled and sequenced on Hiseq X. Raw sequencing reads were trimmed using Trimmomatic-0.32 (ILLUMINACLIP:illumina-adaptors.fa:3:7:7:1:true, LEADING:20, TRAILING:20, SLIDINGWINDOW:4:20, MINLEN:50) (*Bolger et al., 2014*). Data files were inspected with FastQC v0.11.22. Trimmed reads were subsequently mapped to the reference genome with BWA-MEM (version 0.7.12). The SAM output was converted into BAM format, sorted, and indexed via SAMTOOLS v1.3.1 (*Li et al., 2009*). Filtering and realignment was conducted with PICARD v1.119 and GATK (*McKenna et al., 2010*). Briefly, the reads were relabeled, sorted, and indexed with AddOrReplaceReadGroups. Duplicated reads were marked with the PICARD feature MarkDuplicates and reads were realigned with first creating a target list with RealignerTargetCreator, second by IndelRealigner from the GATK suite. Resulting reads were again sorted and indexed with SAMTOOLS. Next, the guidance for PSMC' (https://github.com/stschiff/msmc/blob/master/guide.md) was followed; VCF-files and masked files were generated with the *bamCaller.py* script (MSMC-tools package). This step requires the chromosome coverage information to mask regions with too low or too high coverage. As recommended in the guidelines, the average coverage per chromosome was calculated using SAMTOOLS. In addition, this step was performed using a coverage threshold of 18 as recommended by *Nadachowska-Brzyska et al., 2016*. Final input data were generated using the *generate_multihetsep.py* script (MSMC-tools package). Subsequently, for each sample PSMC' was run independently. Bootstrapping was performed for 30 samples per individual and input files were generated with the *multihetsep_bootstrap.py* script (MSMCtools package).

## Analysis of differential expressed genes with age

We downloaded the previously published RNAseq data from a longitudinal study of *Nothobranchius furzeri* (*Reichwald et al., 2015*). The data set contains five time points (5 w, 12 w, 20 w, 27 w, 39 w) in three different tissues (liver, brain, skin). The raw reads were mapped to the NFZ v2.0 reference genome and subsequently counted using STAR (version 2.6.0 .c) (*Dobin et al., 2013*) and FeatureCounts (version 1.6.2) (*Liao et al., 2014*). We performed statistical analysis of differential expression with age using DESeq2 (*Love et al., 2014*) and age as factor. Genes are classified as upregulated in

young (log(FoldChange)<0, adjusted p<0.01), upregulated in old (log(FoldChange)>0,adjusted p<0.01).

## Acknowledgements

We would like to thank Patience and Edson Gandiwa for their administrative support, Tamuka Nhi-watiwa for helping with logistics and samples handling; Evious Mpofu, Hugo and Elsabe van der Westhuizen and all the rangers of the Gonarezhou National Park for their support in the field. We are thankful to Zimbabwe National Parks for allowing our team to conduct research in the Gonarez-hou National Park; Itamar Harel, Matej Polacik and Radim Blazek for hands-on contribution with the field work. We further thank all members of the Valenzano lab for their continuous scientific input and support. The Czech Science Foundation provided financial support to MR for sampling Mozam-bican populations (19–01789S). This project was funded by the Max Planck Institute for Biology of Ageing, the Max Planck Society and the CECAD at the University of Cologne.

## Additional information

### Funding

| Funder | Grant reference number | Author |
|---|---|---|
| Max Planck Society | Valenzano Group core funding | David Willemsen Rongfeng Cui Dario Riccardo Valenzano |
| Czech Science Foundation | 19-01789S | Martin Reichard |

The funders had no role in study design, data collection and interpretation, or the decision to submit the work for publication.

### Author contributions

David Willemsen, Resources, Formal analysis, Validation, Investigation, Visualization, Methodology, Writing - review and editing; Rongfeng Cui, Formal analysis; Martin Reichard, Resources; Dario Ric-cardo Valenzano, Conceptualization, Resources, Supervision, Funding acquisition, Investigation, Methodology, Writing - original draft, Project administration, Writing - review and editing

### Author ORCIDs

Rongfeng Cui (iD) http://orcid.org/0000-0002-8146-6958
Martin Reichard (iD) http://orcid.org/0000-0002-9306-0074
Dario Riccardo Valenzano (iD) https://orcid.org/0000-0002-8761-8289

### Decision letter and Author response

Decision letter https://doi.org/10.7554/eLife.55794.sa1
Author response https://doi.org/10.7554/eLife.55794.sa2

## Additional files

### Supplementary files

• Supplementary file 1. file 1A. Geographical and environmental statistics file 1B. Sequencing and mapping statistics file 1C. Genome-wide genetic differentiation file 1D. $F_{ST}$ outlier high genetic dif-ferentiation NF414 vs. NF303 file 1E. $F_{ST}$ outlier high genetic differentiation NF303 vs. GNP file 1F. $F_{ST}$ outlier high genetic differentiation NF414 vs. GNP file 1G. $F_{ST}$ outlier low genetic differentiation file 1H. Distribution of fitness effects file 1I. Pathway overrepresentation DoS file 1J. $F_{ST}$ outlier genes differentially expressed with age file 1K. Pairwise Wilcoxon rank sum test P values for Consurf score with *N. orthonotus* as outgroup file 1L. Pairwise Wilcoxon rank sum test P values for Consurf score with *N. rachovii* as outgroup

• Transparent reporting form

## Data availability

Genome sequencing data have been deposited on GenBank under the following accession code: BioProject ID: PRJNA599375 The pooled data sequences used for population genetics under the following accession code: Bioproject PRJNA627180.

The following datasets were generated:

| Author(s) | Year | Dataset title | Dataset URL | Database and Identifier |
|---|---|---|---|---|
| Cui R, Willemsen D, Valenzano DR | 2020 | Nothobranchius furzeri assembly 2.0 | https://www.ncbi.nlm.nih.gov/bioproject/PRJNA599375/ | NCBI BioProject, PRJNA599375 |
| Willemsen D, Valenzano DR | 2020 | Pooled whole genome sequencing and individual resequencing of wild-caught Nothobranchius furzeri | https://www.ncbi.nlm.nih.gov/bioproject/PRJNA627180/ | NCBI BioProject, PRJNA627180 |

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
