## [Decision Letter]

**Acceptance summary:**

Your study on using turquoise killifish to determine the evolutionary forces shaping life history divergence within species is of great interest to the field of aging. The use of genome sequencing and population genetics allows to test some of the major evolutionary theories of aging. Your findings that small population size and genetic drift determine the accumulation of deleterious gene variants in specific genes leading to short lifespan will have great impact on the field bringing those who think of evolutionary and molecular mechanisms of aging closer.

**Decision letter after peer review:**

Thank you for submitting your article "Intra-Species differences in population size shape life history and genome evolution" for consideration by *eLife*. Your article has been reviewed by three peer reviewers, and the evaluation has been overseen by a Reviewing Editor and Jessica Tyler as the Senior Editor.

The reviewers have discussed the reviews with one another and the Reviewing Editor has drafted this decision to help you prepare a revised submission.

As the editors have judged that your manuscript is of interest, but as described below that additional experiments are required before it is published, we would like to draw your attention to changes in our revision policy that we have made in response to COVID-19 (https://elifesciences.org/articles/57162). First, because many researchers have temporarily lost access to the labs, we will give authors as much time as they need to submit revised manuscripts. We are also offering, if you choose, to post the manuscript to bioRxiv (if it is not already there) along with this decision letter and a formal designation that the manuscript is 'in revision at *eLife*'. Please let us know if you would like to pursue this option. (If your work is more suitable for medRxiv, you will need to post the preprint yourself, as the mechanisms for us to do so are still in development.)

In addition to a point by point rebuttal, in particular please respond to the following points:

1) The major conclusions are based on too narrow an interpretation of the senescence literature. They have not considered mutation accumulation as a basis for their results. They need to do so and to adjust the rest of the paper accordingly or at least need to explain to me why they did not include it.

2) The authors should explain the inferred DFEs and show that the DFE inference method to produce reasonable results on simulated data with a similar demographic history to what they infer from the killifish data.

“There is also concern about the DFE study, since, if I understand Figure 4D correctly, the wetter habitat populations have a higher proportion of strongly deleterious SNPs than the GNP (dry) population. Does this go against the argument presented in the paper?”

Reviewer #1:

In their manuscript, the authors investigate the evolutionary forces shaping lifespan in killifish populations. They produced an improved genome assembly and annotation and generate population genetic data for 3 populations of the species. By contrasting the relative wetness of the habitats and using an array of population genetic analyses, they make statements about their effective population sizes, loci with signatures of positive selection (extreme differentiation), and contrasts in their distributions of fitness effects (DFEs). They conclude that the QTL loci previously linked to differences in lifespan did not show signs of positive selection and that genetic drift has allowed a relatively large proportion of slightly deleterious polymorphisms to persist in the smaller population with the most arid habitat.

Generally, I found the study interesting and the choice of system an important one since this model is being used for longevity research. The conclusion that lifespan differences between populations is driven mainly by drift has implications for how results from this model are interpreted in the future. There are a few major concerns which I believe are related to this conclusion. They are detailed below.

1) The study ultimately cannot comment on the initial ancestral evolution of shorter time to reproductive maturity or lifespan because it is analyzing differences between populations, all of which already have short lifespans and time to maturity. This is not a critical concern however, and the authors seems to be aware of this, but it should be stated upfront so readers appreciate the distinction. It remains a possibility that the species-wide trait of short lifespan and time to reproductive maturity arose through selection in response to the extreme seasonality of their habitat.

2) It is not entirely clear whether the populations studied here are the same as those in which the lifespan QTL were discovered. What are the genetic and geographic distances between these populations and those studied by Valenzano et al.

If the populations are substantially different, then the finding of no evidence of positive selection at the QTL is limited, since lifespan associated QTL in these populations could be entirely different. This is an important concern to address.

3) The reader would benefit from explanations about the expectations for the α statistic, since many will not have encountered it before. What are expectations under relevant scenarios? What do values of 0, 1, and negative really mean? How robust are such inferences to demographics and other confounders?

Because conclusions of that paper are somewhat reliant on these values and related tests about the DFEs and DoS, it is important to spell this out and to understand their interpretation, strengths and weaknesses.

4) Each of the population genetic tests could be affected by demographic changes. The authors should address this possibility head on and explain how the method or its interpretation is or is not affected by the scenarios they inferred in Figure 1A.

5) The DFE analysis in Figure 4D seems to suggest a very high proportion of highly deleterious SNPs. ~60% of SNPs with a value of 4Nes < -100. This value seems high. Wouldn't we expect most segregating SNPs to be nearly neutral?

Could the authors explain? Is this high proportion of highly deleterious SNPs due to SNPs at very low frequency?

6) Given point 5 above, the wet population has a much higher proportion of highly deleterious SNPs than the arid population. Couldn't it be argued that the wet population is suffering poorer genetic health, since it has a higher proportion of highly damaging SNPs?

The Consurf analysis seems more convincing to me than the DFE analysis.

Reviewer #2:

The authors present a diversity of analyses of patterns of genetic variation among populations of the annual killifish *Nothobranchus fuerzi*. These fish live in ponds that are dry for large portions of every year. They are able to persist because they lay eggs that are desiccation resistant, but also because the adults have very rapid development, enabling them to mature quickly and reproduce during the brief times that the ponds hold water. The adults also age rapidly and have short life expectancies. The different populations represented in the study come from ponds distributed along a gradient of annual rainfall, so some ponds are destined to retain water for much shorter intervals of time than others. The fish from ponds which are expected to have short durations do not have faster development in the laboratory, which is surprising, yet they do have faster aging and shorter life expectancies. This paper is one of a series in which the authors argue that the real reason these fish have shorter life expectancies is that these ponds of smaller effective population sizes and hence are more susceptible to genetic drift.

In offering this review, I must also acknowledge my background and limitations. I am familiar with the biology of these fish, with the empirical study of aging and with the literature on the evolution of aging. I also have a good background in evolutionary biology. I am not a practitioner of the sorts of genetic methods employed in this study and am less able to judge them.

I have two categories of concern about this paper. The first is that I feel it is poorly written and inappropriately written for the general audience targeted by a journal like e*Life*. Second, I feel that there are critical deficiencies in the science.

Presentation: This work is actually a test of alternative hypotheses, yet the authors never clearly state this. They instead present an argument in favor of one of the two hypotheses while the contrast between the two is implicit. The two hypotheses are: 1) the differences in life expectancy among the various populations are a consequence of how lifespan evolved in each of them, 2) the shorter lifespans of fish from ponds of shorter duration are a by-product of genetic drift.

Hypothesis 1: There are diverse hypotheses for the evolution of lifespan and aging, but I will only address three of them here. The two that date to the 1950's are "antagonistic pleiotropy" (Williams, 1957) and "mutation accumulation" (Medawar, 1952). Medawar's hypothesis is relevant here, since he predicts that species/populations that experience shorter life expectancy will more readily accumulate deleterious mutations that act late in life because natural selection (purifying selection) will weaken with age as a consequence of the small number of individuals that survive to advanced ages.

In supporting their argument for the importance of genetic drift, the authors show an accumulation of genetic variation in the population that presumably has reduced life expectancy plus they present evidence that this variation is likely to be deleterious. A problem I see with their argument is that mutation accumulation makes the same prediction, save for the restriction that the deleterious mutations be ones that are age-specific in their effects. The fact is that they do not know whether or not the variation they characterize is age-specific in its expression. I cannot see how they can discriminate between deleterious mutations that accumulate because of genetic drift versus those that accumulate because of mutation accumulation. The only compelling argument I do see is their evidence for effective population size, but the low end of the estimated population size for the more rapidly aging population is in the vicinity of 20,000, which is not that small and not by itself likely to generate severe genetic drift. Furthermore, this population has been declining in size over time. My understanding of the expected trend for long term declines in population size (based on the application of this idea to whales by Steve Palumbi) is that there will be more genetic variation than expected, given the current population size.

A second implicit argument for genetic drift is the results of their 2017 paper in Evolution in which they quantify other aspects of the life histories of multiple populations of this species. This paper is a lab study of the life histories of multiple populations of a number of different species of annual killifish, one of which is *N. fuerzi*. The study is good in terms of the spectrum of life history variables they assess. A noteworthy feature of the results for *N. fuerzi* is the differences among populations in lifespan in a controlled laboratory environment (populations from ponds with shorter expected duration also tend to have shorter lifespans) without any clear evidence of a tradeoff in other aspects of the life history (e.g., those populations that are shorter lived also mature at an earlier age and/or invest more in offspring). In fact, the short-lived population did tend to invest more in eggs early in life, but the difference was relatively small. As an aside, they put some effort into addressing a third hypothesis for the evolution of aging, which is condition-dependent selection (Williams and Day, 2003). I think that condition dependence is an unlikely mechanism in this system. It can be effective if short life expectancy is due to predation, for example, because adaptation to predation can indirectly affect aging. When a pool dries, there is no option for adaptation in a fish – they all die.

We can agree that there is not strong support for any form of antagonistic pleiotropy revealed in the 2017 paper. The best shot is a slightly higher level of reproductive investment in the population from a shorter-lived pond, which would balance against reduced lifespan. I would not feel tempted to make this argument, but I would advise not banking too heavily on this study fully eliminating antagonistic pleiotropy as a source of differences among populations in lifespan. I would be inclined to first take another shot at this approach.

In summary, they need to be explicit about the alternative hypotheses and need to address the possible role of mutation accumulation. Note that there is empirical evidence in favor of a contribution of mutation accumulation to aging. In many other regards, their presentation is not one that will readily appeal to a general audience. I found their figures to be particularly difficult to interpret.

Science: I have confounded presentation with science a bit since my argument about mutation accumulation applies to the scientific aspects of the presentation, but there are other issues as well. I summarize some of these in the following comments on specific portions of the manuscript.

Other comments:

Subsection “Population genetics of natural turquoise killifish populations”: If they are working with lab lines, then how can we know whether any genetic drift associated with the different lines represents differences among natural populations vs. drift that occurred during prolonged laboratory culture. The effective size of the laboratory lines and their duration in captivity matter. If the genomes were just from wild-caught fish then this is not an issue, but it could apply to the results reported in the 2017 Evolution paper.

Another issue is to better illustrate the climatic differences among the study populations. I can see that they fall at different places in a rainfall gradient, but it would be better to give a more concrete comparison, like the duration of holding water. Are any such comparisons available?

Subsection “Genetic differentiation among turquoise killifish populations”: Are positive natural selection and purifying selection the only interpretations for the F_ST_ values? I had thought that various forms of balancing selection can yield the same results that you attribute to purifying selection (with regard to purifying selection – shouldn't this interpretation be broadened to include balancing selection?).

This section is a bit deceptive in presentation. These regions are all ones that were not associated with any QTL for lifespan, correct? This means that you are looking for age-specific differences in expression in the absence of any affiliation with lifespan, save one exception. I understand the bottom line, which is that you will suggest that the absence of any positive evidence for a genetic basis in lifespan the alternative of genetic drift becomes more plausible. You can do a much better job of making these alternatives more explicit. *But*, there are very good reasons why there could be portions of the genome under selection that were not revealed by your QTL study. Consider adding a power analysis of these results if you really want to make this argument; e.g., how big could an effect have been yet not show up as significant? What if the evolution of aging and lifespan were polygenic, with each gene making a small contribution? Positive results from QTL can be very informative, but I do not thing the reverse is true.

Delete subsection “Evolutionary origin of the sex chromosome”. It has nothing to do with the central theme of the paper, so it is an unnecessary digression. It also impressed me as a topic that could be expanded on and published on its own.

Subsection “Relaxed selection in turquoise killifish populations”: I understand the goal of this section of the paper, which is to make a compelling case for genetic drift as the cause of differences among populations in lifespan. This is the part of the paper that I am not well qualified to judge so I hope other reviewers can do better. But, I can at least question whether or not mutation accumulation could be a plausible alternative, since I believe it is consistent with the patterns reported here. If it is not, then you need to take it on then explain why not.

Subsection “Relaxation of selection in age-related disease pathways”: My reaction to this section is the same as the last section. I am limited in my ability to evaluate this, but I also question whether mutation accumulation represents a viable alternative explanation.

Discussion paragraph two: Eliminate these lines and save them for a later paper on the sex chromosome.

Discussion: I think that the Discussion must be redone in concert with the Introduction. The Introduction should formally present the alternative explanations for differences among populations in aging – antagonistic pleiotropy, mutation accumulation, drift. I think it is okay to just refer to the 2017 paper to dismiss other alternatives, like condition dependent selection. I do not see how you can discriminate between drift and mutation accumulation. I am particularly concerned by your reporting a reasonably large effective population size for the population from the driest environment, in spite of its being the smallest population. If you really want to go this route, then I think it is incumbent on you to do some simulations, or at least something concrete, to argue that the population size is small enough for drift to have been an issue. These issued would be subjects for the Discussion.

Subsection “Koeppen-Geiger index and bioclimatic variables”: This is an obscure way of reporting differences in local climate and pond longevity. It would make a big difference if you could give me something more concrete than this since the values you report look small and subtle but the implied effects on biology are large. The two do not fit well together.

Reviewer #3:

This paper presents a new killifish genome assembly as well as population sequencing data from several killifish populations living in wet and dry habitats. Using this dataset, the authors show that killifish lifespan is inversely correlated with several measures of population diversity. They find no evidence for positive selection favoring genetic variation that appears to shorten lifespan based on a prior QTL analysis, but instead find support for the idea that increased genetic drift could have caused lifespan-shortening variation to reach high frequency in small, outlying killifish populations. Overall the study is very nice and provides some compelling evidence that killifish lifespan has been shortened by weakly deleterious mutation accumulation.

1) My one comprehensive issue with the manuscript is that it implies too strongly that the dryness of the outlying killifish habitats directly caused those populations to lose diversity and accumulate deleterious mutations. Even if all killifish habitats had identical rainfall and resources, we would still expect the populations at the origin of the range expansion to have higher diversity and less genetic load than smaller, more recently founded, outlying groups. In the absence of any data from control pairs of killifish populations that are geographically distant from each other but have similar levels of rainfall, the paper should clearly state up front that the differences in diversity between populations might not be driven primarily by wetness vs dryness but would be expected to exist whenever one population is more recently founded than another. In addition to addressing this broad issue, it would be helpful for the authors to address the following more specific issues:

2) The manuscript notes that F_ST_ outlier genes are differentially expressed between fish of different ages but does not comment on how often random control genes likewise vary in expression as a function of age.

3) For the gene with unusually low F_ST_ that is located on the sex chromosome, is its outlier status appropriately calibrated with respect to the overall divergence and diversity levels of the sex chromosomes, which can differ substantially from that of the autosomes?

4) In this discussion of the origin of the killifish sex chromosome, it wasn't clear to me whether the sex determining locus was inherited ancestrally or seems to have acquired a de novo sex determining function.

5) How good is the evidence that the DFE inference method being used is robust to non-equilibrium demographic history?

[Editors' note: further revisions were suggested prior to acceptance, as described below.]

Thank you for submitting your article "Intra-Species differences in population size shape life history and genome evolution" for consideration by *eLife*. Your article has been reviewed by three peer reviewers, and the evaluation has been overseen by a Reviewing Editor and Jessica Tyler as the Senior Editor. The reviewers have opted to remain anonymous.

The reviewers have discussed the reviews with one another and the Reviewing Editor has drafted this decision to help you prepare a revised submission.

In this study the authors use turquoise killifish to determine the evolutionary forces shaping life history divergence within species. Their results support that mutation accumulation plays an important role in shaping lifespan, but the reviewers have also raised concerns that must be addressed before acceptance of this manuscript.

Essential revisions:

1) The authors still do not state how close these populations are to those studied in the QTL study (Kirschner et al., 2012). Couldn't they just provide coordinates, a map, physical distances, or something to help the reader understand how close they are? As I previously stated, if they are not close, then it raises issues of whether these populations would even have the same loci affecting differences in lifespan. The authors provide the new statements below, but it is not clear how to interpret them objectively. Were these fish collected meters from the previous ones? Kilometers? 100 km?

"The populations used in this study are from localities in proximity to those used in the previous QTL study."

"we collected wild populations from natural localities as close as possible to the locations of origin of the strains used for the QTL study."

The revision also adds:

"…or that the populations used in this study and those used for the QTL analysis had a different genetic architecture of lifespan"

This statement could be unnecessary if exact locations show them to be quite close.

2) The explanation of MK *α* is still inadequate. Some PopGen aficionados will know, but general *eLife* readers won't know what these values mean. What are high values, low values… etc. What has changed in the sequences to cause these values to change? The authors must really present these statistics in a didactic way.

3) From the new explanations of the inferred DFE of new mutations, it seems this analysis only reflects inferred population demographics, rather than the actual deleteriousness of polymorphisms in the population. Is it correct to say that it uses inferred demographics to just rescale a statistical distribution chosen to represent the DFE?

If so, why not then just present the inferred demographic history? If the DFE analysis isn't analyzing anything about the actual polymorphisms present in the populations, why make that extra inferential jump? Just report the demographics. If this analysis reflects some other quality of the data that translates into actual fitness effects, then it should be stated. If it reflects demographics alone as they would affect a theoretical DFE, then that should be clearly stated.

4) The authors have done a thorough job with revisions and substantially improved their manuscript. The one point from my review that I'd like to revisit is my point that more recently founded populations tend to have smaller effective size and less diversity than older populations regardless of founding order. I don't agree with the reasoning that this only applies to temporally separate populations-there is extensive literature in humans on a "serial bottleneck model" that explains why population diversity is proportional to the distance of a population from the origin of the human range expansion in Africa (see e.g. Ramachandran et al., 2005). This work was based on genetic sampling of human populations at the same time, as was done with the killifish populations here, but founding order is important because more recently founded populations have had less time to recover from the associated bottlenecking founder effect. I don't think it's necessary for the authors to do more work to address this point, but I do think they should point out in the text that some proportion of the population size difference between the wet and dry populations is probably due to the fact that killifish have existed at large population size in the wet habitats for a long time, whereas the dry population probably expanded from a small founder population more recently.

---

## [Author Response]

In addition to a point by point rebuttal, in particular please respond to the following points:1) The major conclusions are based on too narrow an interpretation of the senescence literature. They have not considered mutation accumulation as a basis for their results. They need to do so and to adjust the rest of the paper accordingly or at least need to explain to me why they did not include it.

We thank the reviewer to raise this important point. Our findings indeed lend full support to mutation accumulation being the main driver of senescence evolution in our empirical example of natural turquoise killifish populations. Indeed, we believe that our work provides a direct link between the classical theories of ageing to empirical findings in ecology and evolution, connecting the mutation accumulation theory with ecological and demographic constraints (habitat aridity and population size, respectively) in wild species. Our main result is that genetic drift due to small effective population size exacerbates the effects of mutation accumulation. We agree that this was not clearly stated in the text and have now explicitly spelled out in Abstract, Introduction and Discussion.

2) The authors should explain the inferred DFEs and show that the DFE inference method to produce reasonable results on simulated data with a similar demographic history to what they infer from the killifish data.“There is also concern about the DFE study, since, if I understand Figure 4D correctly, the wetter habitat populations have a higher proportion of strongly deleterious SNPs than the GNP (dry) population. Does this go against the argument presented in the paper?”

We fully agree with the editor and with the reviewers that the DFE analysis demands clearer explanation.

We would like to point out that the difference between the DFE spectra between dry and wet populations is along the expected direction. DFE refers to the fitness effects of *newly arising mutations*. When purifying selection is strong, more new mutations will be strongly deleterious and thus be removed effectively from the population. That is, the strongly deleterious mutations are not observed in the data. However, when purifying selection is weaker, some of the deleterious mutations “move” to the weakly deleterious category and segregate in the population. This is why the population with the smaller effective population size (dry population) has less strongly deleterious new mutations, but relatively more slightly deleterious mutations. Another way to see this point is that mutations with same negative selection coefficient *s* would be more deleterious (more to the left in the DFE plot) in larger populations than in smaller ones, as 4*Ne*s scales linearly with population size.

To address this point numerically, we performed a new simulation with SLiM (version 3.3, Haller and Messer, 2019), using similar demographic parameters to those from the PSMC’ analysis. We simulated four models with two populations that diverged from the same ancestral population and are identical in all population genetic parameters. Importantly, the selection coefficient *s* of the deleterious mutations was drawn from the same reflected gamma distribution (mean of -2.5 and shape of 0.35). All models start with an initial burn-in phase of 50000 generations to generate genetic diversity and a stable ancestral population. We re-scaled the population genetic parameters to an initial population size of 10k to increase the speed of the simulation. This should not affect the results and is a common method. The models vary in either having immediate population size changes (Model A, B) or having exponential growth (Model C, D). We further distinguished between a split directly after the burn-in phase (Model B, D) or at a later timepoint, following the PSMC’ analysis interpretation (Model A, C).

The output of the simulation was used to build the site-frequency-spectra that were used as the input for PolyDFE. We found that the PolyDFE results recovered the same conclusions as those we empirically observed between the large and the small population. In fact, although the selection coefficient for the simulations was drawn from the same gamma distribution for large and small populations, the small population evolved fewer strongly deleterious novel mutations, but more slightly deleterious mutations, as expected under weaker purifying selection. The results of this simulation are displayed in Figure 4—figure supplement 1.

Reviewer #1:[…]1) The study ultimately cannot comment on the initial ancestral evolution of shorter time to reproductive maturity or lifespan because it is analyzing differences between populations, all of which already have short lifespans and time to maturity. This is not a critical concern however, and the authors seems to be aware of this, but it should be stated upfront so readers appreciate the distinction. It remains a possibility that the species-wide trait of short lifespan and time to reproductive maturity arose through selection in response to the extreme seasonality of their habitat.

As correctly noted by the reviewer, the current study specifically focuses on population differences within the model species *Nothobranchius furzeri*. Certainly, since all *Nothobranchius* share common adaptations, including embryonic diapause and rapid developmental time, we cannot use the current dataset to search for genes important for genus-wide adaptations that emerged in the common ancestor of *Nothobranchius*. To note, different species in the genus *Nothobranchius* differ in lifespan – with species living up to 2 years – even though they have embryonic diapause and reach sexual maturity roughly at the same time. Similarly, different wild populations of the species *Nothobranchius furzeri* reach sexual maturation at the same time, even though have different lifespan in nature and in captivity. Hence, early life history adaptations (embryonic diapause and rapid sexual maturation) appear somehow disjointed in the genus *Nothobranchius* and in the species *Nothobranchius furzeri*. We believe that African killifish are an example of how early life history traits may be under different selective constraints and evolutionary trajectories than late life history traits. This very aspect was addressed in a previous work conducted by our group and published in 2019 (Cui et al., 2019). The present work is intended to give insights to the mechanisms that resulted in lifespan differences between populations of the turquoise killifish. We now clarify in the text that our study does not address the evolution of rapid sexual maturation and diapause in killifish, but rather provide an evidence of how nearly neutral evolution adds genome-wide deleterious gene variants that predominantly affect late life history traits.

2) It is not entirely clear whether the populations studied here are the same as those in which the lifespan QTL were discovered. What are the genetic and geographic distances between these populations and those studied by Valenzano et al.If the populations are substantially different, then the finding of no evidence of positive selection at the QTL is limited, since lifespan associated QTL in these populations could be entirely different. This is an important concern to address.

This point is well taken. Although the strains used in Valenzano et al. are derived from wild populations that differ in their lifespan, we cannot assume that the lifespan QTL architecture is shared across any pair of long- vs. short-lived turquoise killifish populations. This very question is what we wanted to explore when we first started this study, and therefore we collected wild populations from natural localities as close as possible to the locations of origin of the strains used for the QTL study. We clarify this point now in results.

3) The reader would benefit from explanations about the expectations for the α statistic, since many will not have encountered it before. What are expectations under relevant scenarios? What do values of 0, 1, and negative really mean? How robust are such inferences to demographics and other confounders?Because conclusions of that paper are somewhat reliant on these values and related tests about the DFEs and DoS, it is important to spell this out and to understand their interpretation, strengths and weaknesses.

We fully agree with the reviewer and have now added explanations of what each of these statistics means, what are the expectations, and how our results fit such expectations.

4) Each of the population genetic tests could be affected by demographic changes. The authors should address this possibility head on and explain how the method or its interpretation is or is not affected by the scenarios they inferred in Figure 1A.

PolyDFE fully accounts for demographic changes (population size is in the x axis of each plot). Fluctuations in demography are cancelled out when computing DoS and MK *α*, a desirable feature of these statistics. Indeed, the main argument of our work is that demographic changes, affecting drift, lead to differences in genetic load. Rather than a confounder, we believe that demographic changes are a direct cause of the observed differences among turquoise killifish populations.

5) The DFE analysis in Figure 4D seems to suggest a very high proportion of highly deleterious SNPs. ~60% of SNPs with a value of 4Nes < -100. This value seems high. Wouldn't we expect most segregating SNPs to be nearly neutral?Could the authors explain? Is this high proportion of highly deleterious SNPs due to SNPs at very low frequency?

We realize the DFE analysis needed more explanation and we are thankful to the reviewer for requesting more clarity on this part. DFE computes the fitness effect of novel mutations. Mutations with similar (e.g. negative) selection coefficient *s* would have a more deleterious effect in large populations than in small populations, since 4*s*Ne is bigger as Ne gets larger. Hence, DFE interprets how purifying selection purges mutations given their selection coefficient and the population size. We have now explained this part extensively and performed a novel simulation with SLiM3 to model how DFE differs in a small and a large population evolved from the same ancestral population. The results of this simulation support our findings and interpretations and are now presented as Figure 4—figure supplement 1. We thank again the reviewer for this note, as we feel that our finding was strengthened by the simulation.

6) Given point 5 above, the wet population has a much higher proportion of highly deleterious SNPs than the arid population. Couldn't it be argued that the wet population is suffering poorer genetic health, since it has a higher proportion of highly damaging SNPs?

See response to point 5). More novel mutations are strongly deleterious in the wet population, and thus they are removed very effectively by purifying selection.

The Consurf analysis seems more convincing to me than the DFE analysis.

We appreciate that the reviewer judges that the Consurf analysis is convincing, and we hope that now also the expanded explanation of the DFE analysis strengthens our findings.

Reviewer #2:[…[I have two categories of concern about this paper. The first is that I feel it is poorly written and inappropriately written for the general audience targeted by a journal like eLife. Second, I feel that there are critical deficiencies in the science.Presentation: This work is actually a test of alternative hypotheses, yet the authors never clearly state this. They instead present an argument in favor of one of the two hypotheses while the contrast between the two is implicit. The two hypotheses are: 1) the differences in life expectancy among the various populations are a consequence of how lifespan evolved in each of them, 2) the shorter lifespans of fish from ponds of shorter duration are a by-product of genetic drift.Hypothesis 1: There are diverse hypotheses for the evolution of lifespan and aging, but I will only address three of them here. The two that date to the 1950's are "antagonistic pleiotropy" (Williams, 1957) and "mutation accumulation" (Medawar, 1952). Medawar's hypothesis is relevant here, since he predicts that species/populations that experience shorter life expectancy will more readily accumulate deleterious mutations that act late in life because natural selection (purifying selection) will weaken with age as a consequence of the small number of individuals that survive to advanced ages.In supporting their argument for the importance of genetic drift, the authors show an accumulation of genetic variation in the population that presumably has reduced life expectancy plus they present evidence that this variation is likely to be deleterious. A problem I see with their argument is that mutation accumulation makes the same prediction, save for the restriction that the deleterious mutations be ones that are age-specific in their effects. The fact is that they do not know whether or not the variation they characterize is age-specific in its expression. I cannot see how they can discriminate between deleterious mutations that accumulate because of genetic drift versus those that accumulate because of mutation accumulation.

We thank the reviewer for emphasizing that our manuscript requires a language that is more geared towards a general audience and that we need to be more explicit about the hypotheses that are tested. We hope we now address both these fundamental aspects.

Indeed, our finding that reduced population size in populations from dry environments leads to genome-wide accumulation of deleterious mutations fully echoes Medawar’s mutation accumulation. Our statement is that genetic drift exacerbates the effects of mutation accumulation and that, based on a population genetics approach, we do not find strong support for antagonistic pleiotropism being the main driving evolutionary mechanism leading to short lifespan in killifish populations from dry (hence more ephemeral) environments. Furthermore, in previous work done in our group (Cui et al., 2019), we showed that in annual killifish, genes with higher expression in early life (e.g. developmental genes) are under stronger purifying selection – i.e. accumulate fewer deleterious alleles – than those expressed in late life. In the present work, studying the shortest-known living killifish (turquoise killifish), which is a unique example of species having populations that naturally differ in wild and captive lifespan, we used population genetics to identify the evolutionary forces that have moulded the genome. Indeed, we find that small population size is strictly associated with the genome-wide expansion of slightly deleterious gene variants. Since our main conclusion was not clearly stated in the text, we have now explicitly added key sentences in Abstract, Introduction and Discussion.

The only compelling argument I do see is their evidence for effective population size, but the low end of the estimated population size for the more rapidly aging population is in the vicinity of 20,000, which is not that small and not by itself likely to generate severe genetic drift. Furthermore, this population has been declining in size over time. My understanding of the expected trend for long term declines in population size (based on the application of this idea to whales by Steve Palumbi) is that there will be more genetic variation than expected, given the current population size.

The “small population size” of the population from drier habitat has to be considered relative to the populations from more wet habitats. On a relative scale, smaller populations are expected to experience higher drift. We refrain from making a statement on the absolute scale of population size and drift.

A second implicit argument for genetic drift is the results of their 2017 paper in Evolution in which they quantify other aspects of the life histories of multiple populations of this species. This paper is a lab study of the life histories of multiple populations of a number of different species of annual killifish, one of which is N. fuerzi. The study is good in terms of the spectrum of life history variables they assess. A noteworthy feature of the results for N. fuerzi is the differences among populations in lifespan in a controlled laboratory environment (populations from ponds with shorter expected duration also tend to have shorter lifespans) without any clear evidence of a tradeoff in other aspects of the life history (e.g., those populations that are shorter lived also mature at an earlier age and/or invest more in offspring). In fact, the short-lived population did tend to invest more in eggs early in life, but the difference was relatively small. As an aside, they put some effort into addressing a third hypothesis for the evolution of aging, which is condition-dependent selection (Williams and Day, 2003). I think that condition dependence is an unlikely mechanism in this system. It can be effective if short life expectancy is due to predation, for example, because adaptation to predation can indirectly affect aging. When a pool dries, there is no option for adaptation in a fish – they all die.We can agree that there is not strong support for any form of antagonistic pleiotropy revealed in the 2017 paper. The best shot is a slightly higher level of reproductive investment in the population from a shorter-lived pond, which would balance against reduced lifespan. I would not feel tempted to make this argument, but I would advise not banking too heavily on this study fully eliminating antagonistic pleiotropy as a source of differences among populations in lifespan. I would be inclined to first take another shot at this approach.

We thank the reviewer for the input. We now cite these studies in our Discussion. Indeed, we are unable to reject antagonistic pleiotropy, even though we could not find good support for it. This may be partly due to the difficulty of detecting positive selection from genomic data in general.

In summary, they need to be explicit about the alternative hypotheses and need to address the possible role of mutation accumulation. Note that there is empirical evidence in favor of a contribution of mutation accumulation to aging. In many other regards, their presentation is not one that will readily appeal to a general audience. I found their figures to be particularly difficult to interpret.Science: I have confounded presentation with science a bit since my argument about mutation accumulation applies to the scientific aspects of the presentation, but there are other issues as well. I summarize some of these in the following comments on specific portions of the manuscript.Other comments:Subsection “Population genetics of natural turquoise killifish populations”: If they are working with lab lines, then how can we know whether any genetic drift associated with the different lines represents differences among natural populations vs. drift that occurred during prolonged laboratory culture. The effective size of the laboratory lines and their duration in captivity matter. If the genomes were just from wild-caught fish then this is not an issue, but it could apply to the results reported in the 2017 Evolution paper.

We would like to clarify that the samples used in this study were all wild caught.

Another issue is to better illustrate the climatic differences among the study populations. I can see that they fall at different places in a rainfall gradient, but it would be better to give a more concrete comparison, like the duration of holding water. Are any such comparisons available?

We agree that the duration of holding water is an important factor and would be a very valuable resource. Satellite data from NASA indeed showed that the duration of holding water is longer in the populations in Mozambique compared to the population in Zimbabwe, but the raw data is not accessible to us. We are confident that the provided environmental data is sufficient to illustrate the differences in the habitat of the investigated samples. Furthermore, a paper published by one of the authors in 2008 (Terzibasi et al., 2008), used as a proxy for (inverse) duration of holding water in this region, the ratio of monthly evaporation over precipitation.

Subsection “Genetic differentiation among turquoise killifish populations”: Are positive natural selection and purifying selection the only interpretations for the F_ST_ values? I had thought that various forms of balancing selection can yield the same results that you attribute to purifying selection (with regard to purifying selection – shouldn't this interpretation be broadened to include balancing selection?).

We thank the reviewer for requesting this clarification. Balancing selection is predicted to cause low genetic differentiation between the populations in most of the cases (e.g. Brandt et al., 2017). Only if none of the alleles are shared by both populations, balancing selection can cause a high F_ST_ values. Importantly, balancing selection causes a higher genetic diversity compared to the background genomic diversity. We could not find elevated pairwise nucleotide diversity in the F_ST_ outlier regions, which would support balancing selection. We now included this in the text.

This section is a bit deceptive in presentation. These regions are all ones that were not associated with any QTL for lifespan, correct? This means that you are looking for age-specific differences in expression in the absence of any affiliation with lifespan, save one exception. I understand the bottom line, which is that you will suggest that the absence of any positive evidence for a genetic basis in lifespan the alternative of genetic drift becomes more plausible. You can do a much better job of making these alternatives more explicit. But, there are very good reasons why there could be portions of the genome under selection that were not revealed by your QTL study. Consider adding a power analysis of these results if you really want to make this argument; e.g., how big could an effect have been yet not show up as significant? What if the evolution of aging and lifespan were polygenic, with each gene making a small contribution? Positive results from QTL can be very informative, but I do not thing the reverse is true.

We agree that we cannot be certain that positive selected genes would show up in the previous QTL results. However, our population genetic analysis showed overall little evidence for positive selection. We now discuss the possibility of other explanations in subsection “Genetic differentiation among turquoise killifish populations”.

Delete subsection “Evolutionary origin of the sex chromosome”. It has nothing to do with the central theme of the paper, so it is an unnecessary digression. It also impressed me as a topic that could be expanded on and published on its own.

We thank the reviewer for showing general interest in the findings, despite questioning the suitability of this result to this study. We agree that this part does not belong to the relaxed selection part of the study, but as the sex determining region is under low genetic divergence between the populations and we provide a new general genome resource, we believe that this result provides a valuable addition to this paper.

Subsection “Relaxed selection in turquoise killifish populations”: I understand the goal of this section of the paper, which is to make a compelling case for genetic drift as the cause of differences among populations in lifespan. This is the part of the paper that I am not well qualified to judge so I hope other reviewers can do better. But, I can at least question whether or not mutation accumulation could be a plausible alternative, since I believe it is consistent with the patterns reported here. If it is not, then you need to take it on then explain why not.

We thank the reviewer for explicitly stating the main implication of our study, which is that mutation accumulation is fully compatible with our findings. Indeed, our results fully support mutation accumulation as a major force shaping genome-wide distribution of deleterious gene variants. We now state this several times through the revised manuscript.

Subsection “Relaxation of selection in age-related disease pathways”: My reaction to this section is the same as the last section. I am limited in my ability to evaluate this, but I also question whether mutation accumulation represents a viable alternative explanation.

Mutation accumulation is indeed supported.

Discussion paragraph two: Eliminate these lines and save them for a later paper on the sex chromosome.Discussion: I think that the Discussion must be redone in concert with the Introduction. The Introduction should formally present the alternative explanations for differences among populations in aging – antagonistic pleiotropy, mutation accumulation, drift. I think it is okay to just refer to the 2017 paper to dismiss other alternatives, like condition dependent selection. I do not see how you can discriminate between drift and mutation accumulation.

We amended the Introduction and Discussion and included the hypotheses of evolution of ageing. We think that mutation accumulation and drift work hand in hand. When drift increases, slightly deleterious mutations are more likely to become highly frequent. Our study links genetic drift to the mutation accumulation theory of ageing, combining the nearly neutral theory of molecular evolution with the mutation accumulation theory of ageing. We now make this point clearer throughout the revised manuscript.

I am particularly concerned by your reporting a reasonably large effective population size for the population from the driest environment, in spite of its being the smallest population. If you really want to go this route, then I think it is incumbent on you to do some simulations, or at least something concrete, to argue that the population size is small enough for drift to have been an issue. These issued would be subjects for the Discussion.

The level of drift is always relative to the large population. It is not the absolute level of drift per se that matters, but the proportion of bad alleles that becomes effectively neutral that matters. Our study highlights that the difference in population size between the population could be plausible for the accumulation of deleterious mutations in the dry population. To address this part, we performed several simulations with SLiM and added Figure 4—figure supplement 1.

Subsection “Koeppen-Geiger index and bioclimatic variables”: This is an obscure way of reporting differences in local climate and pond longevity. It would make a big difference if you could give me something more concrete than this since the values you report look small and subtle but the implied effects on biology are large. The two do not fit well together.

The Koeppen-Geiger is the most widely used climate classification system. We believe that monthly precipitation over evapotranspiration is a reasonable way to infer water persistence. However, we understand that these measures are no perfect replacement for measured water persistence.

Reviewer #3:[…]1) My one comprehensive issue with the manuscript is that it implies too strongly that the dryness of the outlying killifish habitats directly caused those populations to lose diversity and accumulate deleterious mutations. Even if all killifish habitats had identical rainfall and resources, we would still expect the populations at the origin of the range expansion to have higher diversity and less genetic load than smaller, more recently founded, outlying groups. In the absence of any data from control pairs of killifish populations that are geographically distant from each other but have similar levels of rainfall, the paper should clearly state up front that the differences in diversity between populations might not be driven primarily by wetness vs dryness but would be expected to exist whenever one population is more recently founded than another. In addition to addressing this broad issue, it would be helpful for the authors to address the following more specific issues:

We thank the reviewer for requiring clarification on these important aspects.

In a previous study, we have indeed shown a robust phylogenetic correlation between climatic parameters and annual life cycle across different killifish species (Cui et al., 2019). Furthermore, our previous study showed in two independent dry-wet pairs of *Nothobranchius* species (*N. orthonotus* and *N. rachovii*) that populations from dry habitats have higher genetic load. We hypothesize that dry habitats lead to population fragmentation (smaller ponds), hence creating a constellation of small, isolated populations. We now emphasize this point further in Discussion.

We understand the point about founding order, but we believe that this reasoning better applies to populations temporally separated, i.e. with more recently founded populations recently diverging from the ancestral population. However, in our case we sampled populations from dry and wet habitats at the same time. We consider these as divergent populations, i.e. derived from the same ancestral population. Sampling two populations simultaneously prevents us from treating any of the sampled populations as the ancestor, as their divergence time from the ancestral population is identical. Thus, according to this model, genetic differences can occur on either branch. Because in the analysis we were making a relative comparison between two contemporary populations, we believe that founding order won’t be applicable. However, as discussed above, smaller populations in dry regions at the margin of the habitat distribution of the species may experience extreme genetic drift due to population fragmentation, contributing to weakening of purifying selection and accumulation of deleterious gene variants. To note, populations from dry regions are also at higher altitude than populations from wet regions. Hence, gene flow is expected to happen from dry to wet, and not vice versa. This, in turn, may further contribute to genetic diversity depletion in dry areas and maintenance of higher diversity in wet regions.

2) The manuscript notes that F_ST_ outlier genes are differentially expressed between fish of different ages but does not comment on how often random control genes likewise vary in expression as a function of age.

We fully agree with this point. We now performed this additional analysis and we couldn’t find an enrichment of significant differentially expressed genes in the F_ST_ outliers. We reported the statistics in the text and still report which genes are significant.

3) For the gene with unusually low F_ST_ that is located on the sex chromosome, is its outlier status appropriately calibrated with respect to the overall divergence and diversity levels of the sex chromosomes, which can differ substantially from that of the autosomes?

We thank the reviewer for this important point. We now checked the range of the F_ST_ values on all chromosomes. We found that the F_ST_ values on the sex chromosome (chromosome 3) were neither the lowest nor the highest values of all chromosomes in all comparisons. Here we attach a boxplot of Z-transformed F_ST_ values for every chromosome and every comparison we did.

**Author response image 1. sa1fig1:** 

4) In this discussion of the origin of the killifish sex chromosome, it wasn't clear to me whether the sex determining locus was inherited ancestrally or seems to have acquired a de novo sex determining function.

With our analysis we want to show that the current sex chromosome of the turquoise killifish emerged from two autosomes. Unfortunately, we cannot say whether the sex determining locus was inherited ancestrally. However, since the sex-determining gene occurs in a region corresponding to an ancestral chromosomal translocation, it is likely that it may have acquired its current function simultaneously with this chromosomal rearrangement. However, we cannot yet date the timing of this event.

5) How good is the evidence that the DFE inference method being used is robust to non-equilibrium demographic history?

The program polyDFE can account for demography and polarization error (i.e. knowledge of the ancestral state). We included runs with these options enabled in the analysis of polyDFE. We also simulated SFS (site frequency spectra) using the demography extracted from the PSMC’ results to validate that demography is accounted for and that polyDFE is precise. The simulated data recovered the reflected gamma distribution and confirmed that polyDFE worked for our model. We now present these results as Figure 4—figure supplement 1.

[Editors' note: further revisions were suggested prior to acceptance, as described below.]

Essential revisions:1) The authors still do not state how close these populations are to those studied in the QTL study (Kirschner et al., 2012). Couldn't they just provide coordinates, a map, physical distances, or something to help the reader understand how close they are? As I previously stated, if they are not close, then it raises issues of whether these populations would even have the same loci affecting differences in lifespan. The authors provide the new statements below, but it is not clear how to interpret them objectively. Were these fish collected meters from the previous ones? Kilometers? 100 km?"The populations used in this study are from localities in proximity to those used in the previous QTL study.""we collected wild populations from natural localities as close as possible to the locations of origin of the strains used for the QTL study."The revision also adds:"…or that the populations used in this study and those used for the QTL analysis had a different genetic architecture of lifespan"This statement could be unnecessary if exact locations show them to be quite close.

We have now provided the exact locations of the populations used in the present and in the QTL work as supplementary Figure 1—figure supplement 1. Additionally, we have specified that these populations belong to the same drainage system. The map shows the physical distance among localities. We have referred to this map in the text and spelled out the relative distance in Km in the figure caption.

2) The explanation of MK α is still inadequate. Some PopGen aficionados will know, but general eLife readers won't know what these values mean. What are high values, low values… etc. What has changed in the sequences to cause these values to change? The authors must really present these statistics in a didactic way.

We have now expanded the section relative to the McDonald-Kreitman statistics to clarify this metrics to a broader audience. We now explain the implications of this metrics and how the plots should be interpreted. I hope this expanded part clarifies better the utility of using this powerful approach.

3) From the new explanations of the inferred DFE of new mutations, it seems this analysis only reflects inferred population demographics, rather than the actual deleteriousness of polymorphisms in the population. Is it correct to say that it uses inferred demographics to just rescale a statistical distribution chosen to represent the DFE?If so, why not then just present the inferred demographic history? If the DFE analysis isn't analyzing anything about the actual polymorphisms present in the populations, why make that extra inferential jump? Just report the demographics. If this analysis reflects some other quality of the data that translates into actual fitness effects, then it should be stated. If it reflects demographics alone as they would affect a theoretical DFE, then that should be clearly stated.

DFE indeed accounts for both demography and deleteriousness of the variants. Hence, it is not redundant to the inferred demographic history. When inferring the DFE, demography (which is inferred from "neutral"/synonymous sites in the genome) needs to be statistically accounted for, because the pattern of genetic variation observed in the data changes due to past demographic history. However, the inferred DFE itself reflects the deleterious or beneficial newly arising functional mutations (nonsynonymous sites). Hence, DFE takes into consideration the actual polymorphisms present in the populations at neutral and functionally relevant sites, from which it estimates actual fitness effects. By showing how different populations with different demographic history have a different distribution of fitness effects of newly emerging mutations, we do feel that the DFE analysis adds a significant novelty to our results, connecting demography with fitness. We clearly stated in the text that DFE is measured based on polymorphisms “To directly estimate the fitness effect of gene variants associated with each population, we analyzed population-specific genetic polymorphisms to assign mutations as beneficial, neutral or detrimental, and determine the distribution of fitness effect (DFE)^42^ of new mutations”, hence we are probably unable to better address this point raised by the reviewer.

4) The authors have done a thorough job with revisions and substantially improved their manuscript. The one point from my review that I'd like to revisit is my point that more recently founded populations tend to have smaller effective size and less diversity than older populations regardless of founding order. I don't agree with the reasoning that this only applies to temporally separate populations-there is extensive literature in humans on a "serial bottleneck model" that explains why population diversity is proportional to the distance of a population from the origin of the human range expansion in Africa (see e.g. Ramachandran et al., 2005). This work was based on genetic sampling of human populations at the same time, as was done with the killifish populations here, but founding order is important because more recently founded populations have had less time to recover from the associated bottlenecking founder effect. I don't think it's necessary for the authors to do more work to address this point, but I do think they should point out in the text that some proportion of the population size difference between the wet and dry populations is probably due to the fact that killifish have existed at large population size in the wet habitats for a long time, whereas the dry population probably expanded from a small founder population more recently.

We thank the reviewer for the feedback on our revised manuscript. We also agree with the statement that founding order and population isolation may play a key role in explaining the differences in population size that we observe among populations. Indeed, in the Introduction we mention the importance of population fragmentation in dry populations and we now have added in Discussion a mention to the possible isolation of populations from drier habitats. However, we still know too little about the temporal sequence of population separation. Our favorite hypothesis though is that indeed populations from more coastal areas may be more ancestral. We have now added in Discussion a part referring to how limited gene flow in isolated populations may be also caused by recent founder effect.